# FLOWER: A FLOW-MATCHING SOLVER FOR INVERSE PROBLEMS

**Mehrsa Pourya**   **Bassam El Rawas**   **Michael Unser**
Biomedical Imaging Group, EPFL
Lausanne, Switzerland
{mehrsa.pourya, bassam.elrawas, michael.unser}@epfl.ch

## ABSTRACT

We introduce *Flower*, a solver for linear inverse problems. It leverages a pre-trained flow model to produce reconstructions that are consistent with the observed measurements. *Flower* operates through an iterative procedure over three steps: (i) a flow-consistent destination estimation, where the velocity network predicts a denoised target; (ii) a refinement step that projects the estimated destination onto a feasible set defined by the forward operator; and (iii) a time-progression step that re-projects the refined destination along the flow trajectory. We provide a theoretical analysis that demonstrates how *Flower* approximates Bayesian posterior sampling, thereby unifying perspectives from plug-and-play methods and generative inverse solvers. On the practical side, *Flower* achieves state-of-the-art reconstruction quality while using nearly identical hyperparameters across various linear inverse problems. Our code is available at https://github.com/mehrsapo/Flower.

## 1 INTRODUCTION

Inverse problems are central to computational imaging and computer vision (McCann & Unser, 2019; Zeng, 2001). Their goal is to reconstruct an underlying signal $\mathbf{x} \in \mathbb{R}^d$ from its observed measurements $\mathbf{y} \in \mathbb{R}^M$. Here, we focus on linear inverse problems, such that the acquisition of the measurements follows the model

$$\mathbf{y} = \mathbf{H}\mathbf{x} + \mathbf{n} \tag{1}$$

for some linear forward operator $\mathbf{H} \colon \mathbb{R}^d \to \mathbb{R}^M$ and additive white Gaussian noise $\mathbf{n} \sim \mathcal{N}(\mathbf{0}, \sigma_n^2 \mathbf{I})$. From a Bayesian perspective, the simplest reconstruction approach is to obtain the maximum-likelihood estimation

$$\hat{\mathbf{x}}_{\mathrm{MLE}} = \arg\max_{\mathbf{x} \in \mathbb{R}^d} p_{\mathbf{Y}|\mathbf{X}=\mathbf{x}}(\mathbf{y}) = \arg\min_{\mathbf{x} \in \mathbb{R}^d} \frac{1}{2\sigma_n^2} \|\mathbf{H}\mathbf{x} - \mathbf{y}\|_2^2. \tag{2}$$

However, this problem is ill-posed and yields poor-quality solutions. Another approach is to obtain the maximum a posteriori estimation (MAP)

$$\hat{\mathbf{x}}_{\mathrm{MAP}} = \arg\max_{\mathbf{x} \in \mathbb{R}^d} p_{\mathbf{X}|\mathbf{Y}=\mathbf{y}}(\mathbf{x}) = \arg\min_{\mathbf{x} \in \mathbb{R}^d} \left( \frac{1}{2\sigma_n^2} \|\mathbf{H}\mathbf{x} - \mathbf{y}\|_2^2 - \log p_{\mathbf{X}}(\mathbf{x}) \right), \tag{3}$$

which requires the knowledge of the prior distribution $p_{\mathbf{X}}$ of images, a quantity that is generally unknown. The minimization problem of equation 3 is consistent with the variational perspective of inverse problems, where the term $(-\log p_{\mathbf{X}}(\mathbf{x}))$ is replaced by a regularizer $\mathcal{R}(\mathbf{x})$ that encodes some properties of the images. From classic signal processing to the advent of deep learning, the design of a good regularizer $\mathcal{R}$ has been of interest. Classic signal processing relies on the smoothness or sparsity of images to introduce wavelet- or total-variation-based regularizers (Rudin et al., 1992; Figueiredo & Nowak, 2003; Beck & Teboulle, 2009). Some methods build upon classical models and try to learn such criteria in a data-driven manner (Roth & Black, 2009; Goujon et al., 2024; Ducotterd et al., 2025; Pourya et al., 2025). Plug-and-play (PnP) approaches focus on the implicit replacement of $\mathcal{R}$ by its proximal operator, with a learned neural network that serves as a denoiser

(Venkatakrishnan et al., 2013; Zhang et al., 2022; Hurault et al., 2022b;a). Although MAP estimations tend to have a good reconstruction quality, they do not necessarily provide the minimum-mean-square estimator $\hat{\mathbf{x}}_{\mathrm{MMSE}}$ that is best in terms of the peak signal-to-noise ratio (PSNR). To estimate $\hat{\mathbf{x}}_{\mathrm{MMSE}}$, one would have to compute the posterior mean $\hat{\mathbf{x}}_{\mathrm{MMSE}} = \mathbb{E}[\mathbf{X}|\mathbf{Y} = \mathbf{y}]$. Moreover, for perceptual metrics, it is better to generate a sample from $p_{\mathbf{X}|\mathbf{Y}=\mathbf{y}}$ instead of an estimator of the distribution.

The objective of generative modeling is to sample from a target distribution $p_{\mathbf{X}}$. In practice, this distribution is unknown, and one typically only has access to a finite collection of its samples. Numerous approaches have been proposed to address this issue. Among them, diffusion models (Sohl-Dickstein et al., 2015; Ho et al., 2020; Song et al., 2020) and, more recently, flow-matching methods (Lipman et al., 2023) represent the state of the art in scalable generative modeling for images.

Flow matching, introduced by Lipman et al. (2023), constructs a continuous-time generative process by parameterizing the velocity field of an ordinary differential equation (ODE) as a neural network. It takes inspiration from optimal transport and continuous normalizing flows (Ambrosio et al., 2008; Hagemann et al., 2022) and transports an initial source distribution $p_{\mathbf{X}_0}$ to a target distribution $p_{\mathbf{X}_1} \approx p_{\mathbf{X}}$. The choice of probability paths from $p_{\mathbf{X}_0}$ to $p_{\mathbf{X}_1}$ are numerous, with Gaussian paths recovering diffusion as a special case (Albergo & Vanden-Eijnden, 2023). However, flow matching mostly focuses on straight-line paths, which yields competitive performance and improved sampling efficiency (Liu et al., 2023; Liu, 2022).

The remarkable success of generative models in image generation motivates their extension to inverse problems, where the goal shifts from the sampling of the prior distribution $p_{\mathbf{X}_1}$ to the sampling of the posterior $p_{\mathbf{X}_1|\mathbf{Y}=\mathbf{y}}$. Several inverse solvers based on diffusion models have been introduced (Chung et al., 2023; 2024; Kawar et al., 2022; Song et al., 2023; Zhu et al., 2023; Zhang et al., 2025; Mardani et al., 2024). Recent efforts also focus on flow-based solvers (Pokle et al., 2024; Martin et al., 2025). Existing approaches can be broadly grouped into two categories: (i) methods that approximate the posterior score (velocity field) with gradient corrections along the generative path; and (ii) PnP strategies that alternate between generative (diffusion or flow) updates and data-consistency steps.

In this work, we introduce a novel solver based on flow matching that achieves state-of-the-art results among flow-based methods for linear inverse problems. Our approach departs from existing methods by framing the problem through a Bayesian ancestral-sampling perspective, which gives rise to a simple three-step procedure with a natural plug-and-play interpretation. Our main contributions are as follows.

1. **Flow-matching solver for inverse problems.** We introduce *Flower*, an inverse problem solver that consists of three steps: (1) a *flow-consistent destination estimation*, where the velocity network is used to predict a destination, interpretable as denoising; (2) a *measurement-aware refinement*, in which the estimated destination is projected onto the feasible set defined by the forward operator; and (3) a *time progression*, where the refined destination is re-projected along the flow path.

2. **Bayesian analysis and relation to PnP.** We provide a Bayesian analysis in which we demonstrate how and under what considerations *Flower* generates approximate posterior samples from $p_{\mathbf{X}_1|\mathbf{Y}=\mathbf{y}}$. Specifically, we show that Step 1 computes the conditional expectation $\mathbb{E}[\mathbf{X}_1|\mathbf{X}_t = \mathbf{x}_t]$, which we then use for the approximation $\tilde{p}_{\mathbf{X}_1|\mathbf{X}_t=\mathbf{x}_t}$ of $p_{\mathbf{X}_1|\mathbf{X}_t=\mathbf{x}_t}$. Through this approximation, we show that Step 2 generates a sample $\tilde{\mathbf{x}}(\mathbf{x}_t, \mathbf{y}) \sim \tilde{p}_{\mathbf{X}_1|\mathbf{X}_t=\mathbf{x}_t,\mathbf{Y}=\mathbf{y}}$. Step 3 then updates the trajectory given the refined destination $\tilde{\mathbf{x}}(\mathbf{x}_t, \mathbf{y})$ and draws a sample $\mathbf{x}_{t+\Delta t} \sim \tilde{p}_{\mathbf{X}_{t+\Delta t}|\mathbf{X}_t=\mathbf{x}_t,\mathbf{Y}=\mathbf{y}}$, which by induction and ancestral sampling, produces the final sample $\mathbf{x}_1 \sim \tilde{p}_{\mathbf{X}_1|\mathbf{Y}=\mathbf{y}}$. These steps rely on three assumptions: the velocity network is optimally trained for unconditional flow matching; the acquisition of measurements follows the forward model in equation 1; and the source and target distributions are independent. To the best of our knowledge, this Bayesian construction is novel within flow-based solvers for inverse problems. Although this construction is key to the derivation of our solver, the resulting procedure closely mirrors the PnP methods. Thus, our Bayesian justification establishes a link between the PnP approach and

approximate posterior sampling with generative models for linear inverse problems. We also discuss a possible extension of *Flower* to nonlinear inverse problems.

3. **Numerical validation.** We first examine a controlled setup with Gaussian mixture models and show that *Flower* successfully recovers posterior samples. We then evaluate our method on standard inverse problem benchmarks for flow matching. We achieve competitive performance, with nearly identical hyperparameters across all tasks.

The remainder of this paper is organized as follows. In Section 2, we review the fundamentals of flow matching along with the mathematical tools required for the development of our method. We then introduce *Flower* in Section 3 and present the associated theoretical analysis. In Section 4, we discuss related work and highlight their similarities and differences with our approach. We report our numerical results in Section 5. Finally, we provide a potential nonlinear extension of *Flower* in Section 6.

## 2 BACKGROUND

### 2.1 FLOW MATCHING

Let $p_{\mathbf{X}_0}$ be a source distribution that is easy to sample and let $p_{\mathbf{X}_1}$ be a target distribution that we want to sample from. A time-dependent flow $\psi_t$ transports $p_{\mathbf{X}_0}$ to $p_{\mathbf{X}_1}$ via the ODE

$$\frac{\mathrm{d}\psi_t(\mathbf{x})}{\mathrm{d}t} = \mathbf{v}_t\big(\psi_t(\mathbf{x})\big), \qquad t \in [0, 1], \tag{4}$$

for some velocity field $\mathbf{v}_t : \mathbb{R}^d \to \mathbb{R}^d$. The intermediate variables $\mathbf{X}_t = \psi_t(\mathbf{X}_0)$ follow a distribution $p_{\mathbf{X}_t}$. The objective of flow matching is to approximate $\mathbf{v}_t$ with a neural network $\mathbf{v}_t^\theta$, which will allow us to sample from $p_{\mathbf{X}_1}$. However, the determination of the flow-matching loss

$$\mathcal{L}_{\mathrm{FM}}(\theta) = \mathbb{E}_{t \sim \mathcal{U}[0,1]} \mathbb{E}_{\mathbf{x}_t \sim p_{\mathbf{X}_t}} \left[ \left\| \mathbf{v}_t^\theta(\mathbf{x}_t) - \mathbf{v}_t(\mathbf{x}_t) \right\|_2^2 \right] \tag{5}$$

is challenging, as it requires access to the marginal velocity field $\mathbf{v}_t(\mathbf{x}_t)$. To address this, we focus on the conditional velocity $\mathbf{v}_t(\mathbf{x}_t \mid \mathbf{x}_1)$ and define the conditional straight-line flow and velocity

$$\mathbf{x}_t = \psi_t(\mathbf{x}_0 \mid \mathbf{x}_1) = (1 - t)\,\mathbf{x}_0 + t\,\mathbf{x}_1, \quad \mathbf{v}_t(\mathbf{x}_t \mid \mathbf{x}_1) = \mathbf{x}_1 - \mathbf{x}_0. \tag{6}$$

This leads to the practical conditional flow-matching loss

$$\mathcal{L}_{\mathrm{CFM}}(\theta) = \mathbb{E}_{t \sim \mathcal{U}[0,1]} \mathbb{E}_{(\mathbf{x}_0, \mathbf{x}_1) \sim \pi} \left[ \left\| \mathbf{v}_t^\theta\big((1 - t)\,\mathbf{x}_0 + t\,\mathbf{x}_1, t\big) - (\mathbf{x}_1 - \mathbf{x}_0) \right\|_2^2 \right], \tag{7}$$

where $\pi \in \Pi(p_{\mathbf{X}_0}, p_{\mathbf{X}_1})$ is a coupling over $(\mathbf{X}_0, \mathbf{X}_1)$, given by joint distributions on $\mathbb{R}^d \times \mathbb{R}^d$ with marginals $p_{\mathbf{X}_0}$ and $p_{\mathbf{X}_1}$. Lipman et al. (2023) have shown that the minimization of $\mathcal{L}_{\mathrm{CFM}}$ is equivalent to the minimization of $\mathcal{L}_{\mathrm{FM}}$, since their gradients with respect to $\theta$ are equal.

The coupling $\pi$ determines how $(\mathbf{x}_0, \mathbf{x}_1)$ are paired. With the *independent* (IND) coupling $\pi = p_{\mathbf{X}_0} \otimes p_{\mathbf{X}_1}$, the training is simple and scalable. However, the resulting interpolated paths can overlap, which may slow down convergence. At the other extreme, the *optimal transport* (OT) coupling $\pi^\star \in \arg\min_{\pi \in \Pi(p_{\mathbf{X}_0}, p_{\mathbf{X}_1})} \mathbb{E}_{(\mathbf{x}_0, \mathbf{x}_1) \sim \pi} \left[ \|\mathbf{x}_1 - \mathbf{x}_0\|_2^2 \right]$ produces globally aligned pairs such that straight-line flows approximate displacement interpolation along the Wasserstein-2 geodesic. If the Monge map $T$ satisfying $T_\# p_{\mathbf{X}_0} = p_{\mathbf{X}_1}$ exists and is known, then no training is needed: the sampling $\mathbf{x}_0 \sim p_{\mathbf{X}_0}$ and the computation of $\mathbf{x}_1 = T(\mathbf{x}_0)$ already generate a sample from $p_{\mathbf{X}_1}$. In practice, $T$ is unknown and its approximation is infeasible. A practical compromise is *mini-batch OT*, which solves an entropically regularized OT problem within each batch to compute an approximate coupling $\hat{\pi}$. This improves alignment over independence with moderate computational overhead. For more details on the mathematical background of OT, such as the definition and uniqueness of the Monge map, we refer to Peyré (2025).

The choice of the source distribution $p_{\mathbf{X}_0}$ is crucial for effective training and sampling. In practice, $p_{\mathbf{X}_0}$ is often chosen as the standard normal distribution $\mathcal{N}(\mathbf{0}, \mathbf{I})$. With the independent coupling, this simply yields $p_{\mathbf{X}_t \mid \mathbf{X}_1 = \mathbf{x}_1} = \mathcal{N}(t\mathbf{x}_1, (1 - t)^2 \mathbf{I})$. However, the computation of $p_{\mathbf{X}_t \mid \mathbf{X}_1 = \mathbf{x}_1}$ in the OT case is challenging due to the mini-batch approach, in which it is difficult to determine the batch a sample $\mathbf{x}_1$ came from, as well as its associated OT paths.

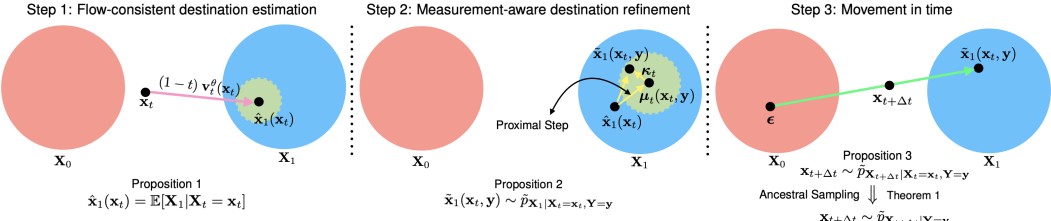

Figure 1: Overview of the three steps in *Flower*. Starting from an initial sample $\mathbf{x}_0 \sim p_{\mathbf{X}_0}$ at time $t$, the method: Step 1 predicts a flow-consistent destination $\hat{\mathbf{x}}_1(\mathbf{x}_t)$; Step 2 refines this destination using the measurements via a proximal step and associated uncertainty sampling to obtain $\tilde{\mathbf{x}}_1(\mathbf{x}_t, \mathbf{y})$; and Step 3 updates the trajectory along time by interpolating $\tilde{\mathbf{x}}_1(\mathbf{x}_t, \mathbf{y})$ with new noise $\boldsymbol{\epsilon} \sim p_{\mathbf{X}_0}$. The $N$-time repetition of these steps yields the final reconstruction $\mathbf{x}_1$.

## 2.2 PROXIMAL OPERATOR

The *proximal operator* of a proper, lower semi-continuous convex function $f \colon \mathbb{R}^d \to \mathbb{R} \cup \{+\infty\}$ is defined as

$$\mathrm{prox}_f(\mathbf{x}) = \underset{\mathbf{w} \in \mathbb{R}^d}{\arg\min} \left( \frac{1}{2} \|\mathbf{w} - \mathbf{x}\|_2^2 + f(\mathbf{w}) \right). \tag{8}$$

This operator can be interpreted as a generalized projection of $\mathbf{x}$ onto a set associated with $f$, balancing proximity to $\mathbf{x}$ and regularization by $f$. Proximal operators play a central role in optimization algorithms that solve inverse problems and are key components of proximal-gradient methods (Bubeck, 2015).

## 3 METHOD

Let $\mathbf{v}_t^\theta$ denote a velocity network trained to generate samples from $p_{\mathbf{X}_1}$ through flow matching. Therefore, starting from $\mathbf{x}_0 \sim p_{\mathbf{X}_0}$, we get a sample $\mathbf{x}_1 \sim p_{\mathbf{X}_1}$ if we perform $N$ iterations of the update equation

$$\mathbf{x}_{t+\Delta t} = \mathbf{x}_t + \Delta t \, \mathbf{v}_t^\theta(\mathbf{x}_t) \tag{9}$$

with $\Delta t = \frac{1}{N}$. We aim to use the pre-trained velocity network $\mathbf{v}_t^\theta$ to generate solutions $\mathbf{x}_1$ that are consistent with the flow and the linear forward model $\mathbf{y} = \mathbf{H}\mathbf{x}_1 + \mathbf{n}$ for $\mathbf{n} \sim \mathcal{N}(\mathbf{0}, \sigma_n^2 \mathbf{I})$, as described in equation 1. To achieve this goal, we introduce *Flower* which, given the measurements $\mathbf{y}$, modifies the unconditional flow path of equation 9 and outputs $\mathbf{x}_1$ by iterating $N$ times over three steps. We first introduce these steps and then theoretically establish how and under what assumptions *Flower* approximates a sample $\mathbf{x}_1$ of the conditional posterior $p_{\mathbf{X}_1|\mathbf{Y}=\mathbf{y}}$. The three steps are as follows.

1. **Flow-consistent destination estimation**
$$\hat{\mathbf{x}}_1(\mathbf{x}_t) = \mathbf{x}_t + (1-t)\mathbf{v}_t^\theta(\mathbf{x}_t). \tag{10}$$

2. **Measurement-aware destination refinement**
$$\tilde{\mathbf{x}}_1(\mathbf{x}_t, \mathbf{y}) = \boldsymbol{\mu}_t(\mathbf{x}_t, \mathbf{y}) + \gamma \boldsymbol{\kappa}_t \tag{11}$$

   for
$$\boldsymbol{\mu}_t(\mathbf{x}_t, \mathbf{y}) = \mathrm{prox}_{\nu_t^2 F_{\mathbf{y}}}\left(\hat{\mathbf{x}}_1(\mathbf{x}_t)\right), \quad \boldsymbol{\kappa}_t \sim \mathcal{N}(\mathbf{0}, \boldsymbol{\Sigma}_t), \tag{12}$$

   where $F_{\mathbf{y}}(\mathbf{x}) = \frac{1}{2\sigma_n^2} \|\mathbf{H}\mathbf{x} - \mathbf{y}\|_2^2$ and $\mathrm{prox}_{\nu_t^2 F_{\mathbf{y}}}$ denotes the proximal operator of $\nu_t^2 F_{\mathbf{y}}$ as defined in equation 8. We have that $\nu_t = \frac{(1-t)}{\sqrt{t^2+(1-t)^2}}$ and that $\boldsymbol{\Sigma}_t = \left(\nu_t^{-2}\mathbf{I} + \sigma_n^{-2}\mathbf{H}^\top\mathbf{H}\right)^{-1}$. The hyperparameter $\gamma \in \{0, 1\}$ controls the consideration of the uncertainty of the destination refinement step.

3. **Movement in time**
$$\mathbf{x}_{t+\Delta t} = (1 - t - \Delta t)\boldsymbol{\epsilon} + (t + \Delta t)\tilde{\mathbf{x}}_1(\mathbf{x}_t, \mathbf{y}), \tag{13}$$

   where $\boldsymbol{\epsilon}$ is newly sampled from $p_{\mathbf{X}_0}$ at each iteration.

Here, $\Delta t = \frac{1}{N}$ and the scheme is initialized with a sample $\mathbf{x}_0 \sim p_{\mathbf{X}_0}$. In Figure 1, we present a visual illustration of these three steps. We also summarize these steps in Algorithm 1 of the Appendix.

We now interpret *Flower* through a Bayesian lens. We assume that, at each iteration, the three steps collectively draw $\mathbf{x}_{t+\Delta t}$ from the transition distribution $p_{\mathbf{X}_{t+\Delta t}|\mathbf{X}_t=\mathbf{x}_t,\mathbf{Y}=\mathbf{y}}$. Under this assumption and with proper initialization, the procedure performs ancestral sampling along the conditional trajectory. By induction, we obtain $\mathbf{x}_{t+\Delta t} \sim p_{\mathbf{X}_{t+\Delta t}|\mathbf{Y}=\mathbf{y}}$. We formalize this in Theorem 1, with proof in Appendix 8.2.1, which in turn implies that the final sample $\mathbf{x}_1$ produced by *Flower* follows the desired posterior $p_{\mathbf{X}_1|\mathbf{Y}=\mathbf{y}}$. We then detail how, in practice, the three steps realize a draw from $\tilde{p}_{\mathbf{X}_{t+\Delta t}|\mathbf{X}_t=\mathbf{x}_t,\mathbf{Y}=\mathbf{y}}$, which serves as an approximation of $p_{\mathbf{X}_{t+\Delta t}|\mathbf{X}_t=\mathbf{x}_t,\mathbf{Y}=\mathbf{y}}$.

**Theorem 1.** *Let* $\mathbf{x}_0$ *be a sample from* $p_{\mathbf{X}_0|\mathbf{Y}=\mathbf{y}}$. *If* $\mathbf{x}_t$ *is a sample from* $p_{\mathbf{X}_t|\mathbf{Y}=\mathbf{y}}$, *then the sample* $\mathbf{x}_{t+\Delta t}$ *from* $p_{\mathbf{X}_{t+\Delta t}|\mathbf{X}_t=\mathbf{x}_t,\mathbf{Y}=\mathbf{y}}$ *follows* $p_{\mathbf{X}_{t+\Delta t}|\mathbf{Y}=\mathbf{y}}$.

**Remark 1.** *For the inductive argument to hold,* Flower *must be initialized with a sample from the conditional distribution* $p_{\mathbf{X}_0|\mathbf{Y}=\mathbf{y}}$. *When* $\mathbf{X}_0$ *and* $\mathbf{X}_1$ *are assumed to be independent, this reduces to a sampling from the unconditional prior* $p_{\mathbf{X}_0}$, *which is often chosen as* $\mathcal{N}(\mathbf{0},\mathbf{I})$.

Theorem 1 presupposes the existence of an ancestral-sampling scheme to generate samples from $p_{\mathbf{X}_{t+\Delta t}|\mathbf{Y}=\mathbf{y}}$. This scheme requires a sampling from the transition distribution $p_{\mathbf{X}_{t+\Delta t}|\mathbf{X}_t=\mathbf{x}_t,\mathbf{Y}=\mathbf{y}}$. We now describe how to realize this transition in practice. We proceed sequentially and explain the details of each step of *Flower*.

First, under the assumption that $\mathbf{v}_t^\theta$ is the optimal velocity network, we show in Proposition 1 that the predicted $\hat{\mathbf{x}}_1(\mathbf{x}_t)$ in Step 1 equals the conditional expectation $\mathbb{E}[\mathbf{X}_1 \mid \mathbf{X}_t = \mathbf{x}_t]$. The proof is provided in Appendix 8.2.2.

**Proposition 1.** *If* $\mathbf{v}^\theta(\mathbf{x}_t, t) = \mathbf{v}_t^*(\mathbf{x})$ *is a pre-trained velocity vector field that minimizes the conditional flow-matching loss, then*

$$\hat{\mathbf{x}}_1(\mathbf{x}_t) = \mathbb{E}[\mathbf{X}_1|\mathbf{X}_t = \mathbf{x}_t] = \mathbf{x}_t + (1-t)\mathbf{v}^\theta(\mathbf{x}_t, t). \tag{14}$$

Since the distribution $p_{\mathbf{X}_1|\mathbf{X}_t=\mathbf{x}_t}$ is not directly available, we propose to approximate it with

$$\tilde{p}_{\mathbf{X}_1|\mathbf{X}_t=\mathbf{x}_t} = \mathcal{N}(\hat{\mathbf{x}}_1(\mathbf{x}_t), \nu_t^2\mathbf{I}), \tag{15}$$

an isotropic Gaussian distribution centered at $\hat{\mathbf{x}}_1(\mathbf{x}_t) = \mathbb{E}[\mathbf{X}_1|\mathbf{X}_t = \mathbf{x}_t]$ with a time-varying covariance. As $t \to 1$, the distribution $p_{\mathbf{X}_t}$ approaches the target $p_{\mathbf{X}_1}$. For $\tilde{p}_{\mathbf{X}_1|\mathbf{X}_t=\mathbf{x}_t}$ to be consistent with this property, $\nu_t$ should anneal in time. We choose $\nu_t = (1-t)/\sqrt{t^2 + (1-t)^2}$, which results in the valid covariance when $p_{\mathbf{X}_1}$ is a standard Gaussian distribution. Our approximation is indeed the $\Pi$GDM approximation proposed by Song et al. (2023) within diffusion solvers and later by Pokle et al. (2024) for flow matching. However, instead of having a score-based interpretation and using this approximation to obtain $\nabla_{\mathbf{x}_t} \log \tilde{p}_{\mathbf{Y}|\mathbf{X}_t=\mathbf{x}_t}$, we propose to sample $\tilde{\mathbf{x}}_1(\mathbf{x}_t, \mathbf{y})$ from the distribution $\tilde{p}_{\mathbf{X}_1|\mathbf{X}_t=\mathbf{x}_t,\mathbf{Y}=\mathbf{y}}$ that approximates $p_{\mathbf{X}_1|\mathbf{X}_t=\mathbf{x}_t,\mathbf{Y}=\mathbf{y}}$. To this end, we show in Proposition 2 that $\tilde{p}_{\mathbf{X}_1|\mathbf{X}_t=\mathbf{x}_t,\mathbf{Y}=\mathbf{y}}$ is indeed a Gaussian distribution, using the $\Pi$GDM approximation and the forward model of equation 1. The proof is provided in Appendix 8.2.3.

**Proposition 2.** *Suppose that* $\tilde{p}_{\mathbf{X}_1|\mathbf{X}_t=\mathbf{x}_t} = \mathcal{N}(\hat{\mathbf{x}}_1(\mathbf{x}_t), \nu_t^2\mathbf{I})$ ($\Pi$GDM approximation) *and* $p_{\mathbf{Y}|\mathbf{X}_1=\mathbf{x}_1} = \mathcal{N}(\mathbf{H}\mathbf{x}_1, \sigma_n^2\mathbf{I})$ *(measurement operation). Then,* $\tilde{p}_{\mathbf{X}_1|\mathbf{X}_t=\mathbf{x}_t,\mathbf{Y}=\mathbf{y}} = \mathcal{N}(\boldsymbol{\mu}_t(\mathbf{x}_t, \mathbf{y}), \boldsymbol{\Sigma}_t)$, *where*

$$\boldsymbol{\mu}_t(\mathbf{x}_t, \mathbf{y}) = \left(\nu_t^{-2}\mathbf{I} + \sigma_n^{-2}\mathbf{H}^\top\mathbf{H}\right)^{-1}\left(\nu_t^{-2}\hat{\mathbf{x}}_1(\mathbf{x}_t) + \sigma_n^{-2}\mathbf{H}^\top\mathbf{y}\right), \tag{16}$$

$$\boldsymbol{\Sigma}_t = \left(\nu_t^{-2}\mathbf{I} + \sigma_n^{-2}\mathbf{H}^\top\mathbf{H}\right)^{-1}. \tag{17}$$

Proposition 2 allows us to sample from $\tilde{p}_{\mathbf{X}_1|\mathbf{X}_t=\mathbf{x}_t,\mathbf{Y}=\mathbf{y}}$ as provided in Step 2 of *Flower* using the re-parameterization trick in equation 11. However, the $\boldsymbol{\mu}_t(\mathbf{x}_t, \mathbf{y})$ in Step 2 (see equation 12) is described using a proximal operator which differs from equation 16. It is easy to verify the equivalence between the two, through the fact that $\boldsymbol{\mu}_t(\mathbf{x}_t, \mathbf{y})$ of equation 16 can be written as the solution to the minimization problem

$$\min_{\mathbf{x}\in\mathbb{R}^d} \left(\frac{1}{2\sigma_n^2}\|\mathbf{H}\mathbf{x} - \mathbf{y}\|_2^2 + \frac{1}{2\nu_t^2}\|\mathbf{x} - \hat{\mathbf{x}}_1(\mathbf{x}_t)\|_2^2\right). \tag{18}$$

This directly results in $\boldsymbol{\mu}_t(\mathbf{x}_t, \mathbf{y}) = \text{prox}_{\nu_t^2 F}(\hat{\mathbf{x}}_1(\mathbf{x}_t))$ for $F_\mathbf{y}(\mathbf{x}) = \frac{1}{2\sigma_n^2}\|\mathbf{H}\mathbf{x} - \mathbf{y}\|_2^2$ under the definition of the proximal operator in equation 8. Moreover, the sampling from the anisotropic Gaussian $\mathcal{N}(\mathbf{0}, \boldsymbol{\Sigma}_t)$ is not trivial; however, if we sample two independent $\boldsymbol{\epsilon}_1 \in \mathbb{R}^d$ and $\boldsymbol{\epsilon}_2 \in \mathbb{R}^M$ from standard Gaussian distributions, then we verify in Appendix 8.2.5 that $\boldsymbol{\kappa}_t = \boldsymbol{\Sigma}_t(\nu_t^{-1}\boldsymbol{\epsilon}_1 + \sigma_n^{-1}\mathbf{H}^\top\boldsymbol{\epsilon}_2)$ follows $\mathcal{N}(\mathbf{0}, \boldsymbol{\Sigma}_t)$.

Step 3 of *Flower* aims to sample the distribution $p_{\mathbf{X}_{t+\Delta t}|\mathbf{X}_t=\mathbf{x}_t, \mathbf{Y}=\mathbf{y}}$, which is also what we required for our ancestral-sampling procedure to hold. We now show that if we have a sample $\tilde{\mathbf{x}}_1(\mathbf{x}_t, \mathbf{y})$ from $\tilde{p}_{\mathbf{X}_1|\mathbf{X}_t=\mathbf{x}_t, \mathbf{Y}=\mathbf{y}}$, then we could obtain a sample from the distribution $\tilde{p}_{\mathbf{X}_{t+\Delta t}|\mathbf{X}_t=\mathbf{x}_t, \mathbf{Y}=\mathbf{y}}$ using equation 13. We first compute $\tilde{p}_{\mathbf{X}_{t+\Delta t}|\mathbf{X}_t=\mathbf{x}_t, \mathbf{Y}=\mathbf{y}}$ under the assumption that $p_{\mathbf{X}_0}$ is independent of $p_{\mathbf{X}_1}$ in Proposition 3 which we prove in Appendix 8.2.4.

**Proposition 3.** *If from the pre-trained flow matching we have that $p_{\mathbf{X}_0} = \mathcal{N}(\mathbf{0}, \mathbf{I})$, if $p_{\mathbf{X}_0}$ is independent of $p_{\mathbf{X}_1}$, and if $\tilde{p}_{\mathbf{X}_1|\mathbf{X}_t=\mathbf{x}_t, \mathbf{Y}=\mathbf{y}}(\mathbf{x}_1) = \mathcal{N}(\boldsymbol{\mu}_t(\mathbf{x}_t, \mathbf{y}), \boldsymbol{\Sigma}_t)$, then it holds that*

$$\tilde{p}_{\mathbf{X}_{t+\Delta t}|\mathbf{X}_t=\mathbf{x}_t, \mathbf{Y}=\mathbf{y}} = \mathcal{N}\left((t+\Delta t)\boldsymbol{\mu}_t, (t+\Delta t)^2\boldsymbol{\Sigma}_t + (1-t-\Delta t)^2\mathbf{I}\right). \tag{19}$$

From Proposition 8.2.4 and by the means of the re-parametrization trick, it is easy to verify that

$$\mathbf{x}_{t+\Delta t} = (1-t-\Delta t)\boldsymbol{\epsilon} + (t+\Delta t)\tilde{\mathbf{x}}_1(\mathbf{x}_t, \mathbf{y}) \tag{20}$$

follows $\tilde{p}_{\mathbf{X}_{t+\Delta t}|\mathbf{X}_t=\mathbf{x}_t, \mathbf{Y}=\mathbf{y}}$ given that $\tilde{\mathbf{x}} \sim \mathcal{N}(\boldsymbol{\mu}_t(\mathbf{x}_t, \mathbf{y}), \boldsymbol{\Sigma}_t)$, which happens in Step 3 of *Flower*.

In our Bayesian justification, we assumed that $p_{\mathbf{X}_0}$ and $p_{\mathbf{X}_1}$ are independent. This assumption excludes, for example, the mini-batch optimal-transport coupling. Nevertheless, in practice, *Flower* can still be applied in such settings and interpreted in a PnP manner. The iterative structure of *Flower* closely resembles PnP methods: Step 1 acts as a denoising step, while Step 2 enforces data consistency. From this viewpoint, Step 3 can be seen as a re-projection onto the flow trajectory, as discussed in Martin et al. (2025). However, rather than relying solely on this interpretation, we provide a Bayesian justification of the procedure. This perspective highlights a conceptual link between PnP methods and posterior sampling. We also address this empirically in our numerical results. For exact posterior sampling to hold under our approximations, $\gamma$ should be set to one. Interestingly, in practice we find that the choice $\gamma = 0$ (i.e., ignoring the uncertainty in the destination refinement step) yields a better reconstruction quality. We provide further discussion on this effect with our numerical results in Section 5. Moreover, our framework remains valid for more general noise distributions $\mathbf{n} \sim \mathcal{N}(\mathbf{0}, \mathbf{R}_n)$ where $\mathbf{R}_n$ is any symmetric positive-definite matrix, as detailed in Appendix 8.3.

## 4 RELATED WORKS

An extensive body of work adapts pre-trained diffusion or flow priors to inverse problems by modifying the dynamics to approximate the conditional posterior. We first review diffusion-based solvers, then flow-based ones. Throughout this section, we highlight how *Flower* differs from similar methods. We use the flow-matching notation with source $\mathbf{X}_0$ and target $\mathbf{X}_1$.

Among diffusion solvers, DPS (Chung et al., 2023) approximates $p_{\mathbf{Y}|\mathbf{X}_t=\mathbf{x}_t}$ by $p_{\mathbf{Y}|\mathbf{X}_1=\hat{\mathbf{x}}_1(\mathbf{x}_t)}$, where $\hat{\mathbf{x}}_1(\mathbf{x}_t)$ is the diffusion-based denoised version of $\mathbf{x}_t$, which leads to a gradient correction to the diffusion dynamics. $\Pi$GDM (Song et al., 2023) approximates $\tilde{p}_{\mathbf{X}_1|\mathbf{X}_t=\mathbf{x}_t} = \mathcal{N}(\hat{\mathbf{x}}_1(\mathbf{x}_t), \nu_t^2\mathbf{I})$ for some time-annealing $\nu_t$ and replaces the gradient correction of DPS with a pseudoinverse-based update. *Flower* adopts the same approximation as $\Pi$GDM, but the subsequent steps differ. DDS (Chung et al., 2024) shares the same perspective as DPS but replaces the gradient with a proximal step motivated by a manifold-preserving gradient perspective. DiffPIR (Zhu et al., 2023) arrives at a very similar structure through half-quadratic splitting, alternating proximal data updates with diffusion denoising. Both DDS and DiffPIR are structurally close to *Flower* but, unlike *Flower*, they lack the Bayesian justification that interprets the updates as posterior sampling. DAPS (Zhang et al., 2025) also uses ancestral sampling similar to *Flower*, but instead of directly computing $\tilde{p}_{\mathbf{X}_1|\mathbf{X}_t=\mathbf{x}_t, \mathbf{Y}=\mathbf{y}}$, it applies Langevin updates for its evaluation, which is more computationally demanding but extends naturally to nonlinear inverse problems.

In the flow-matching domain, OT-ODE (Pokle et al., 2024) employs a $\Pi$GDM-based approximation, similar in spirit to that of *Flower*. However, instead of adopting our ancestral sampling scheme, they,

similar to the approach of ΠGDM, approximate the score of the conditional distribution $\tilde{p}_{\mathbf{Y}|\mathbf{X}_t=\mathbf{x}_t}$ in order to construct the new velocity field. Therefore, while our method, *Flower*, relies on the same approximation for $\tilde{p}_{\mathbf{X}_1|\mathbf{X}_t=\mathbf{x}_t}$, our methods act in different ways. FlowDPS (Kim et al., 2025) extends diffusion posterior sampling to the flow models by decomposing the ODE via a flow-version of Tweedie's formula, which differs from the ancestral sampling approach of *Flower*. Flow-Priors (Zhang et al., 2024) tackle inverse problems by reformulating the MAP objective as a sequence of time-dependent MAP subproblems with closed-form evaluations by taking advantage of the velocity network. However, unlike *Flower*, their approach relies on the computation of $\mathrm{Tr}\,\nabla\mathbf{v}_t^\theta$, which is costly. D-Flow (Ben-Hamu et al., 2024) adopts an implicit regularization strategy, replacing the data-fidelity objective $\mathbf{x} \mapsto \|\mathbf{H}\mathbf{x} - \mathbf{y}\|^2$ with a latent loss $\mathbf{z} \mapsto \|\mathbf{H}(f(1,\mathbf{z})) - \mathbf{y}\|^2$, where $f$ is the solution of the flow ODE. The latent loss is non-convex with an implicit regularization effect that prevents convergence to trivial solutions. The optimization is performed by back-propagation through ODE solutions, which is more computationally demanding compared to the steps of *Flower*. FLAIR (Erbach et al., 2025) optimizes a decoupled variational objective with a time-dependent calibration scheme, rather than relying on the Bayesian perspective of *Flower*. PnP-Flow (Martin et al., 2025) introduces a PnP framework that employs the velocity network as a denoiser. Its update steps are similar to *Flower*, but we replace their gradient update with a proximal operation, which leads to improved reconstruction quality. Moreover, *Flower* offers a Bayesian justification of the process while PnP-Flow is purely plug-and-play.

## 5 NUMERICAL RESULTS

Here, we benchmark *Flower* against state-of-the-art flow-based inverse solvers across a range of linear inverse problems. In Appendix 8.4.1, we validate the Bayesian interpretation of *Flower* through a toy experiment with Gaussian mixtures, where ground-truth posterior samples are computable. We also present further numerical results for Fourier sampling and non-isotropic Gaussian noise in Appendix 8.5.7.

We implement *Flower* as described in Algorithm 1 of the Appendix. As a practical note, the computation $\mathbf{\Sigma}_t\mathbf{b}$ for any $\mathbf{b} \in \mathbb{R}^d$ requires inversion $(\nu_t^{-2}\mathbf{I} + \sigma_n^{-2}\mathbf{H}^\top\mathbf{H})^{-1}\mathbf{b}$. Wherever this inversion is required, we instead solve the corresponding linear system $(\nu_t^{-2}\mathbf{I} + \sigma_n^{-2}\mathbf{H}^\top\mathbf{H})\mathbf{z} = \mathbf{b}$ using conjugate gradients (CG) with a maximum of 50 iterations and an $\ell_2$-residual tolerance of $10^{-5}$ and return $\mathbf{z}$ as the solution of the operation. We found the CG implementation sufficiently efficient in practice due to the positive-definite structure of $\nu_t^{-2}\mathbf{I} + \sigma_n^{-2}\mathbf{H}^\top\mathbf{H}$.

### 5.1 BENCHMARK EXPERIMENTS

The goal of this section is to benchmark our method against other flow-matching-based solvers for inverse problems. For fair comparisons, we adopt the benchmark introduced by Martin et al. (2025), which also includes state-of-the-art PnP and diffusion models. We describe the datasets and experimental setup for completeness, present quantitative results in Tables 1 and 2, and provide qualitative examples in Figure 2. Finally, we discuss the key observations, highlighting the performance of *Flower* and its empirical considerations.

We use two datasets for our numerical comparisons. First, we use $(128 \times 128)$ human-face images from Yang et al. (2015), denoted by CelebA. Second, we use resized $(256 \times 256)$ cat images from Choi et al. (2020), denoted by AFHQ-Cat. We normalize all images to the range $[-1, 1]$. We train on the full training sets of both datasets. We tune the hyperparameters of different methods using a validation set. For CelebA, we use 32 images from the dataset's validation split. AFHQ-Cat has no validation split, so following Martin et al. (2025), we construct one by selecting 32 images from the test set and removing them from that set. For reporting the metrics, we use 100 test images from each dataset, a limit imposed by the computational cost of baseline methods (D-Flow and Flow Priors). For faster methods (including *Flower*), we also report metrics on larger datasets consisting of 1000 images for CelebA and 400 for AFHQ, which are presented in Appendix 8.5.6. These extended results show the same performance trends as the 100-image evaluations presented in this section.

We compare *Flower* against flow-matching solvers OT-ODE (Pokle et al., 2024), D-Flow (Ben-Hamu et al., 2024), Flow-Priors (Zhang et al., 2024), and PnP-Flow (Martin et al., 2025), as well as two other baselines: PnP-GS (Hurault et al., 2022b), a state-of-the-art plug-and-play method, and

DiffPIR (Zhu et al., 2023), a diffusion-based inverse solver. All models (except DiffPIR) use the same U-Net backbone (Ronneberger et al., 2015) trained with Mini-Batch OT Flow Matching (Tong et al., 2024) and a Gaussian latent prior. Pre-trained weights for the flow models and PnP-GS are taken from Martin et al. (2025), trained with learning rate $10^{-4}$: on CelebA for 200 epochs (batch size 128) and on AFHQ-Cat for 400 epochs (batch size 64). For *Flower*, we additionally train a variant without latent–target coupling (Flower-IND) using the same hyperparameters, which corresponds to our theoretical setting. While Flower-IND achieves higher performance (see Appendix 8.5.2), we primarily report Flower-OT for consistency with other flow-matching baselines. Training DiffPIR with this backbone proved ineffective due to limited capacity, so following Martin et al. (2025), we adopt a pretrained model Choi et al. (2021) from the DeepInv library (Tachella et al., 2023), originally trained on FFHQ (Karras et al., 2019). This introduces some mismatch but provides the fairest diffusion-based baseline. Note that we use the latest checkpoints from Martin et al. (2025), but our averaging strategy differs from theirs. In Martin et al. (2025), results are reported by grouping four images into one batch and then averaging across 25 such batches. In contrast, we recomputed the results using 100 independent averages over the images themselves. Consequently, our reported numbers differ from those in Martin et al. (2025).

We evaluate performance on five restoration tasks: (i) denoising with Gaussian noise with $\sigma_n = 0.2$; (ii) deblurring with a $61 \times 61$ Gaussian kernel ($\sigma_b = 1.0$ for CelebA, $\sigma_b = 3.0$ for AFHQ-Cat) and additive noise $\sigma_n = 0.05$; (iii) super-resolution ($2\times$ downsampling for CelebA and $4\times$ for AFHQ-Cat, with $\sigma_n = 0.05$); (iv) random inpainting with $70\%$ of pixels removed ($\sigma_n = 0.01$); and (v) box inpainting with a centered $40 \times 40$ mask for CelebA and $80 \times 80$ mask for AFHQ-Cat ($\sigma_n = 0.05$). To report quantitative results, we use peak signal-to-noise ratio (PSNR), structural similarity index measure (SSIM), and learned perceptual image patch similarity (LPIPS). Note that for PSNR and SSIM, higher values indicate better performance, while for LPIPS, lower values are better.

To ensure fair comparisons, we adopt the optimal hyperparameters reported in Martin et al. (2025) for each method, obtained via grid search on the validation set to maximize PSNR. For PnP-Flow, we report two variants: PnP-Flow1 and PnP-Flow5, which apply one and five evaluations of the velocity network per denoising step, respectively. For *Flower*, we follow the same procedure, reporting in Tables 1 and 2 either the output of a single evaluation (Flower1-OT) or the average of five evaluations (Flower5-OT). A key property of *Flower* is that, apart from the number $N$ of iterations and the knowledge of the noise level $\sigma_n$, it uses the same hyperparameters across different inverse problems, unlike other flow models. In particular, aside from $N$, the only hyperparameter of *Flower* is $\gamma$, which controls the uncertainty of the destination refinement. Across all setups, $\gamma = 0$ yields higher reconstruction quality. As discussed in Appendix 8.5.3, the choice $\gamma = 1$ produces samples that appear realistic but requires the averaging of multiple runs to achieve competitive PSNR, whereas $\gamma = 0$ attains better quality with fewer averages. This observation is consistent with our toy experiments, where $\gamma = 0$ encouraged sampling from higher-probability regions. For $N$, we always match the number of steps used by our main competitor, PnP-Flow. We provide further ablation studies on the effect of the number of evaluations for the averaging and different time discretizations within *Flower* in Appendix 8.5.4 and 8.5.5, respectively. The full hyperparameter details are reported in Appendix 8.5.8.

**Key Observations.** On CelebA (Table 1), *Flower* achieves the best or near-best results across all tasks, with clear gains in deblurring and box inpainting. The five-step averaging further improves the results. On AFHQ-Cat (Table 2), *Flower* remains highly competitive and outperforms baselines in deblurring, box inpainting, and random inpainting, while PnP-GS is strongest in denoising. In these tables, bold numbers indicate the best results among single-average results of methods. Underlined numbers indicate the second best. Blue numbers highlight the overall best across all methods. We illustrate in Figure 2 representative reconstructions across denoising, deblurring, super-resolution, and inpainting tasks. Compared to OT-ODE, D-Flow, and Flow-Priors, *Flower* consistently produces fewer artifacts, while also avoiding the over-smoothing often observed in PnP-Flow. These visual trends align with the quantitative results and highlight the robustness of *Flower* across diverse degradations. In Figure 3, we illustrate the solution path of *Flower* for the box inpainting task shown in Figure 2. As expected, Step 1 produces flow-based denoised images, while Step 2 enforces consistency with the measurements. In this specific box-inpainting setup, Step 2 primarily aligns the region outside the box with the measurements and preserves the result of Step 1 inside the box. Step 3 then mixes the refined destination with fresh source noise; as $t$ increases, the noise decreases and the reconstruction emerges. The injected noise is essential to prevent the velocity network from

Table 1: Results on 100 test images of the dataset CelebA.

| Method | Denoising | | | Deblurring | | | Super-resolution | | | Random inpainting | | | Box inpainting | | |
|---|---|---|---|---|---|---|---|---|---|---|---|---|---|---|---|
| | PSNR | SSIM | LPIPS | PSNR | SSIM | LPIPS | PSNR | SSIM | LPIPS | PSNR | SSIM | LPIPS | PSNR | SSIM | LPIPS |
| Degraded | 20.00 | 0.348 | 0.372 | 27.83 | 0.740 | 0.126 | 10.26 | 0.183 | 0.827 | 11.95 | 0.196 | 1.041 | 22.27 | 0.742 | 0.214 |
| PnP-GS | **32.64** | 0.910 | 0.035 | 34.03 | 0.924 | 0.041 | 31.31 | 0.892 | 0.064 | 29.22 | 0.875 | 0.070 | - | - | - |
| DiffPIR | 31.20 | 0.885 | 0.060 | 32.77 | 0.912 | 0.060 | 31.52 | 0.895 | 0.033 | 31.74 | 0.917 | 0.025 | - | - | - |
| OT-ODE | 30.54 | 0.859 | **0.032** | 33.01 | 0.921 | 0.029 | 31.46 | 0.907 | **0.025** | 28.68 | 0.871 | 0.051 | 29.40 | 0.920 | 0.038 |
| D-Flow | 26.04 | 0.607 | 0.092 | 31.25 | 0.854 | 0.038 | 30.47 | 0.843 | 0.026 | **33.67** | 0.943 | **0.015** | 30.70 | 0.899 | 0.026 |
| Flow-Priors | 29.34 | 0.768 | 0.134 | 31.54 | 0.858 | 0.056 | 28.35 | 0.713 | 0.102 | 32.88 | 0.871 | 0.019 | 30.07 | 0.858 | 0.048 |
| PnP-Flow1 | 31.80 | 0.905 | 0.044 | 34.48 | 0.936 | 0.040 | 31.09 | 0.902 | 0.045 | 33.05 | **0.944** | 0.018 | 30.47 | 0.933 | 0.037 |
| Flower1-OT (ours) | 32.28 | **0.914** | 0.034 | **34.98** | **0.947** | **0.026** | **32.36** | **0.923** | 0.034 | 33.08 | **0.944** | 0.018 | **31.19** | **0.945** | **0.022** |
| PnP-Flow5 | 32.30 | 0.911 | 0.056 | 34.80 | 0.940 | 0.047 | 31.49 | 0.906 | 0.056 | 33.98 | 0.953 | 0.022 | 31.09 | 0.940 | 0.043 |
| Flower5-OT (ours) | 33.14 | 0.926 | 0.038 | 35.67 | 0.954 | 0.032 | 33.09 | 0.932 | 0.040 | 33.95 | 0.953 | 0.020 | 31.87 | 0.952 | 0.023 |

Table 2: Results on 100 test images of the dataset AFHQ-Cat.

| Method | Denoising | | | Deblurring | | | Super-resolution | | | Random inpainting | | | Box inpainting | | |
|---|---|---|---|---|---|---|---|---|---|---|---|---|---|---|---|
| | PSNR | SSIM | LPIPS | PSNR | SSIM | LPIPS | PSNR | SSIM | LPIPS | PSNR | SSIM | LPIPS | PSNR | SSIM | LPIPS |
| Degraded | 20.00 | 0.314 | 0.509 | 23.94 | 0.517 | 0.444 | 11.70 | 0.208 | 0.873 | 13.36 | 0.223 | 1.081 | 21.80 | 0.740 | 0.198 |
| PnP-GS | **32.58** | **0.894** | **0.072** | 27.91 | 0.753 | 0.349 | 24.15 | 0.632 | 0.362 | 29.42 | 0.836 | 0.126 | - | - | - |
| DiffPIR | 30.58 | 0.835 | 0.189 | 27.56 | 0.728 | 0.342 | 23.65 | 0.624 | 0.402 | 31.70 | 0.881 | 0.062 | - | - | - |
| OT-ODE | 30.03 | 0.815 | 0.076 | 27.06 | 0.713 | **0.123** | 25.91 | 0.716 | **0.108** | 29.40 | 0.839 | 0.090 | 24.62 | 0.875 | 0.085 |
| D-Flow | 26.13 | 0.574 | 0.175 | 27.82 | 0.721 | 0.164 | 24.64 | 0.601 | 0.190 | 32.20 | 0.894 | 0.040 | **26.26** | 0.842 | 0.077 |
| Flow-Priors | 29.41 | 0.763 | 0.153 | 26.47 | 0.700 | 0.181 | 23.51 | 0.570 | 0.272 | 32.37 | 0.906 | 0.047 | 26.20 | 0.818 | 0.118 |
| PnP-Flow1 | 31.18 | 0.863 | 0.135 | 27.87 | 0.760 | 0.304 | **26.94** | **0.763** | 0.171 | **33.00** | **0.918** | **0.037** | 26.00 | 0.897 | 0.103 |
| Flower1-OT (ours) | 31.69 | 0.879 | 0.102 | **28.64** | **0.775** | 0.255 | 26.23 | 0.741 | 0.272 | 32.97 | **0.918** | 0.040 | 26.19 | **0.915** | **0.063** |
| PnP-Flow5 | 31.43 | 0.864 | 0.168 | 28.19 | 0.766 | 0.332 | 27.37 | 0.774 | 0.183 | 33.75 | 0.929 | 0.048 | 26.68 | 0.901 | 0.120 |
| Flower5-OT (ours) | 32.35 | 0.891 | 0.116 | 28.97 | 0.784 | 0.283 | 26.57 | 0.750 | 0.282 | 33.70 | 0.927 | 0.045 | 26.88 | 0.922 | 0.066 |

getting stuck at the previous iterate and to allow it to predict improved destinations. As shown in Table 3 of Appendix 8.5.1, *Flower* has a runtime that is similar to PnP-Flow and OT-ODE, with only a slight overhead relative to PnP-Flow due to the proximal-projection step replacing a simple gradient update, while requiring the same minimal memory. In contrast, D-Flow and Flow-Priors are substantially slower and more memory-intensive.

## 6 POTENTIAL EXTENSION TO NONLINEAR INVERSE PROBLEMS

*Flower* as it is can handle different linear forward operators. To extend it to nonlinear inverse problems, Proposition 2 needs to be revisited. When the measurement operator $\mathbf{H}$ is linear, the likelihood $p_{\mathbf{Y}|\mathbf{X}_1=\mathbf{x}_1} = \mathcal{N}(\mathbf{H}\mathbf{x}_1, \sigma_n^2\mathbf{I})$ is Gaussian, and, combined with the ΠGDM Gaussian prior $\tilde{p}_{\mathbf{X}_1|\mathbf{X}_t=\mathbf{x}_t} = \mathcal{N}(\hat{\mathbf{x}}_1(\mathbf{x}_t), \nu_t^2\mathbf{I})$, the approximate posterior remains Gaussian: $\tilde{p}_{\mathbf{X}_1|\mathbf{X}_t=\mathbf{x}_t, \mathbf{Y}=\mathbf{y}} = \mathcal{N}(\boldsymbol{\mu}_t(\mathbf{x}_t, \mathbf{y}), \boldsymbol{\Sigma}_t)$. This yields the closed-form mean $\boldsymbol{\mu}_t$ and covariance $\boldsymbol{\Sigma}_t$ in Proposition 2.

If the measurement model is nonlinear, i.e.,

$$p_{\mathbf{Y}|\mathbf{X}_1=\mathbf{x}_1} = \mathcal{N}(\mathbf{h}(\mathbf{x}_1), \sigma_n^2\mathbf{I}) \tag{21}$$

for some nonlinear function $\mathbf{h}\colon \mathbb{R}^d \to \mathbb{R}^M$, then it still holds that

$$\tilde{p}_{\mathbf{X}_1|\mathbf{X}_t=\mathbf{x}_t, \mathbf{Y}=\mathbf{y}}(\mathbf{x}_1) \propto \exp\Big(-\frac{1}{2\sigma_n^2}\|\mathbf{y}-\mathbf{h}(\mathbf{x}_1)\|^2 - \frac{1}{2\nu_t^2}\|\mathbf{x}_1 - \hat{\mathbf{x}}_1(\mathbf{x}_t)\|^2\Big), \tag{22}$$

which is no longer Gaussian due to the nonlinearity of $\mathbf{h}$. Nevertheless, sampling from this distribution can be done using iterative sampling schemes, since the score (gradient of the log density) is available in closed form:

$$\nabla_{\mathbf{x}_1} \log \tilde{p}_{\mathbf{X}_1|\mathbf{X}_t=\mathbf{x}_t, \mathbf{Y}=\mathbf{y}}(\mathbf{x}_1) = \sigma_n^{-2} J_{\mathbf{h}}(\mathbf{x}_1)^\top (\mathbf{y} - \mathbf{h}(\mathbf{x}_1)) - \nu_t^{-2}\big(\mathbf{x}_1 - \hat{\mathbf{x}}_1(\mathbf{x}_t)\big), \tag{23}$$

where $\mathbf{J_h}$ denotes the Jacobian of $\mathbf{h}$. This score function could be used to generate samples of $\tilde{p}_{\mathbf{X}_1|\mathbf{X}_t=\mathbf{x}_t, \mathbf{Y}=\mathbf{y}}$ via schemes such as Langevin dynamics which provides a valid substitute for Step 2 of *Flower*, which enables the handling of nonlinear cases without requiring modifications to the remaining steps.

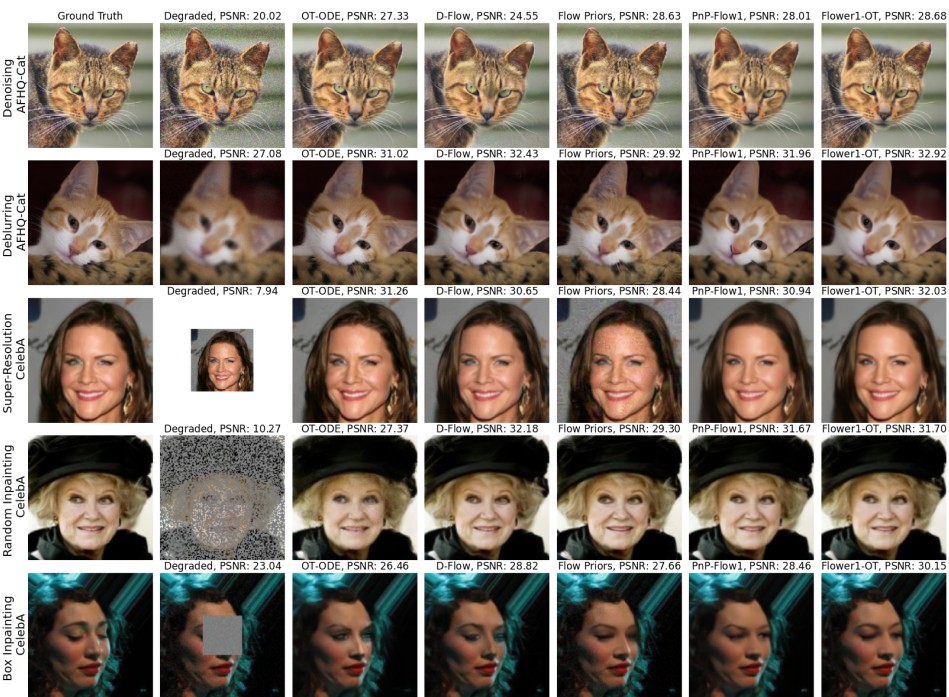

Figure 2: Visual comparison for flow-matching inverse solvers.

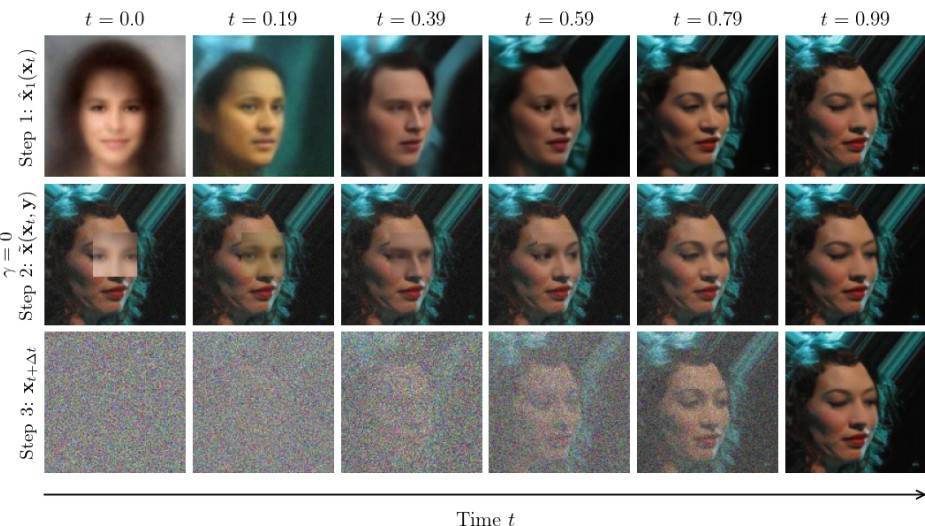

Figure 3: Solution path of *Flower* for box inpainting.

# 7  CONCLUSION

We introduced *Flower*, a method that leverages pre-trained flow-matching models to solve linear inverse problems through a simple three-step iterative procedure. By combining flow-consistent predictions, measurement-aware refinement, and time evolution, *Flower* provides a principled Bayesian interpretation while retaining the plug-and-play flexibility of existing approaches. Our analysis established the conditions under which the method recovers approximate samples from the conditional posterior. Our experiments demonstrated both validity on toy data and state-of-the-art performance across diverse inverse problems.

**Acknowledgements**   The authors acknowledge support from the European Research Council (ERC Project FunLearn, Grant 101020573) and the Swiss National Science Foundation (Grant 200020_219356). We thank Anne Gagneux and Ségolène Martin for their assistance in reproducing results, and Pakshal Bohra for insightful discussions on sampling schemes.

**Ethics Statement.**   This work proposes a methodology for solving inverse problems in imaging using pre-trained generative models. Our method is designed as a general-purpose solver and does not target specific sensitive domains. Our experiments are conducted exclusively on publicly available datasets (CelebA and AFHQ-Cat) that are commonly used in the literature. No private or otherwise sensitive data were collected or used. Our method has potential positive applications in areas such as medical imaging, but, as with other generative techniques, should be applied responsibly to avoid misuse in creating misleading content.

**Reproducibility Statement.**   We aim to ensure that our work is fully reproducible. We provide detailed descriptions of the algorithm (Section 3) and its pseudo-code (Algorithm 1), the theoretical analysis (Appendix 8.2), and the experimental setup (Section 5 and Appendix 8.5), including datasets, hyperparameters, training procedures, and evaluation metrics. All datasets are publicly available and cited in the text, and our method relies on standard architectures and benchmarks, allowing independent verification of our results. Our implementation is available at `https://github.com/mehrsapo/Flower`.

**Use of LLMs.**   The authors of this manuscript acknowledge the use of large language models (LLM) for grammatical polishing and typographic corrections.

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

## 8 APPENDIX

### 8.1 FLOWER ALGORITHM

We outline the steps of *Flower* in Algorithm 1.

---

**Algorithm 1** Flower: Flow Matching Solver for Inverse Problems

---

**Require:** Measurements $\mathbf{y}$, forward operator $\mathbf{H}$, noise level $\sigma_n$, pretrained velocity $\mathbf{v}_t^\theta$, steps $N$, uncertainty flag $\gamma \in \{0, 1\}$
1: Set $\Delta t = 1/N$; sample $\mathbf{x}_0 \sim p_{\mathbf{X}_0}$  $\qquad\qquad\qquad\qquad\qquad\qquad\qquad$ ▷ e.g., $\mathcal{N}(\mathbf{0}, \mathbf{I}_d)$
2: **for** $k = 0$ to $N - 1$ **do**
3: $\qquad t = k\,\Delta t$
4: $\qquad$ **(Step 1) Destination estimate:** $\hat{\mathbf{x}}_1(\mathbf{x}_t) = \mathbf{x}_t + (1 - t)\,\mathbf{v}_t^\theta(\mathbf{x}_t)$
5: $\qquad \nu_t = \frac{1-t}{\sqrt{t^2 + (1-t)^2}}, \quad F_{\mathbf{y}}(\mathbf{x}) = \frac{1}{2\sigma_n^2}\|\mathbf{Hx} - \mathbf{y}\|_2^2$
6: $\qquad$ **(Step 2) Refinement mean:** $\boldsymbol{\mu}_t = \mathrm{prox}_{\nu_t^2 F_{\mathbf{y}}}\big(\hat{\mathbf{x}}_1(\mathbf{x}_t)\big)$
7: $\qquad$ **(Step 2) Optional uncertainty:** $\boldsymbol{\Sigma}_t = \big(\nu_t^{-2}\mathbf{I}_d + \sigma_n^{-2}\mathbf{H}^\top\mathbf{H}\big)^{-1}$
8: $\qquad\qquad$ sample $\boldsymbol{\epsilon}_1 \sim \mathcal{N}(\mathbf{0}, \mathbf{I}_d),\ \boldsymbol{\epsilon}_2 \sim \mathcal{N}(\mathbf{0}, \mathbf{I}_M), \quad \boldsymbol{\kappa}_t = \boldsymbol{\Sigma}_t\big(\nu_t^{-1}\boldsymbol{\epsilon}_1 + \sigma_n^{-1}\mathbf{H}^\top\boldsymbol{\epsilon}_2\big)$
9: $\qquad \tilde{\mathbf{x}}_1(\mathbf{x}_t, \mathbf{y}) = \boldsymbol{\mu}_t + \gamma\,\boldsymbol{\kappa}_t$
10: $\qquad$ **(Step 3) Time progression:** sample $\boldsymbol{\epsilon} \sim p_{\mathbf{X}_0}$ and set
11: $\qquad\qquad \mathbf{x}_{t+\Delta t} = (1 - t - \Delta t)\,\boldsymbol{\epsilon} + (t + \Delta t)\,\tilde{\mathbf{x}}_1(\mathbf{x}_t, \mathbf{y})$
12: **end for**
13: **return** $\mathbf{x}_1$

---

### 8.2 PROOFS

#### 8.2.1 PROOF OF THEOREM 1

To establish this result, we first recall ancestral sampling. It is a procedure that enables us to draw samples from a marginal distribution $p_{\mathbf{Z}_K}$ when the full joint distribution over a sequence of variables $\mathbf{Z} = (\mathbf{Z}_1, \mathbf{Z}_2, \ldots, \mathbf{Z}_K)$ is defined via a chain of conditional densities and when the marginal distribution of $\mathbf{Z}_K$ can be written as

$$p_{\mathbf{Z}_K}(\mathbf{z}_K) = \int_{\mathbf{z}_{K-1}} \cdots \int_{\mathbf{z}_1} p_{\mathbf{Z}_K|\mathbf{Z}_{K-1}=\mathbf{z}_{K-1}}(\mathbf{z}_t) \cdots p_{\mathbf{Z}_2|\mathbf{Z}_1=\mathbf{z}_1}(\mathbf{z}_2)\, p_{\mathbf{Z}_1}(\mathbf{z}_1)\, \mathrm{d}\mathbf{z}_1 \cdots \mathrm{d}\mathbf{z}_{K-1}. \quad (24)$$

Ancestral sampling offers a practical way to generate samples from $\mathbf{p}_{Z_K}$ without direct evaluation of this integral. The process samples sequentially from the distributions

$$\mathbf{z}_1 \sim p_{\mathbf{Z}_1}, \quad \mathbf{z}_2 \sim p_{\mathbf{Z}_2|\mathbf{Z}_1=\mathbf{z}_1}, \quad \ldots, \quad \mathbf{z}_K \sim p_{\mathbf{Z}_K|\mathbf{Z}_{K-1}=\mathbf{z}_{K-1}}. \quad (25)$$

By following this sequence, we obtain a valid sample from the marginal $\mathbf{z}_K \sim p_{\mathbf{Z}_K}$.

*Proof of Theorem 1.* By using the marginal distributions and the general chain rule for joint probability, we obtain

$$p_{\mathbf{X}_{t+\Delta t}|\mathbf{Y}=\mathbf{y}}(\mathbf{x}_{t+\Delta t}) = \int_{\mathbf{x}_t} p_{\mathbf{X}_t, \mathbf{X}_{t+\Delta t}|\mathbf{Y}=\mathbf{y}}(\mathbf{x}_t, \mathbf{x}_{t+\Delta t})\, \mathrm{d}\mathbf{x}_t.$$

$$= \int_{\mathbf{x}_t} p_{\mathbf{X}_t|\mathbf{Y}=\mathbf{y}}(\mathbf{x}_t) p_{\mathbf{X}_{t+\Delta t}|\mathbf{X}_t=\mathbf{x}_t, \mathbf{Y}=\mathbf{y}}(\mathbf{x}_{t+\Delta t})\, \mathrm{d}\mathbf{x}_t. \quad (26)$$

It then suffices to follow the ancestral sampling procedure with $K = 2$, $\mathbf{Z}_1 = \mathbf{X}_t|\mathbf{Y} = \mathbf{y}$, and $\mathbf{Z}_2 = \mathbf{X}_{t+\Delta t}|\mathbf{Y} = \mathbf{y}$ to complete the proof. $\qquad\square$

#### 8.2.2 PROOF OF PROPOSITION 1

*Proof.* Lipman et al. (2023) have shown that the velocity vector field that minimizes the conditional flow-matching loss is

$$\mathbf{v}_t^*(\mathbf{x}) = \mathbb{E}[\mathbf{X}_1 - \mathbf{X}_0|\mathbf{X}_t = \mathbf{x}_t], \quad (27)$$

which then yields

$$\hat{\mathbf{x}}_1(\mathbf{x}_t) = \mathbb{E}[\mathbf{X}_1|\mathbf{X}_t = \mathbf{x}_t] = \mathbb{E}[\mathbf{X}_t + (1-t)(\mathbf{X}_1 - \mathbf{X}_0)|\mathbf{X}_t = \mathbf{x}_t] = \mathbf{x}_t + (1-t)\mathbf{v}_t^\theta(\mathbf{x}). \quad (28)$$

This is the desired result under the assumption that $\mathbf{v}_t^\theta(\mathbf{x}) = \mathbf{v}_t^*(\mathbf{x})$. $\qquad\square$

### 8.2.3 PROOF OF PROPOSITION 2

*Proof.* Using Bayes' rule, we can write

$$p_{\mathbf{X}_1|\mathbf{X}_t=\mathbf{x}_t,\mathbf{Y}=\mathbf{y}}(\mathbf{x}_1) = \frac{p_{\mathbf{Y}|\mathbf{X}_1=\mathbf{x}_1,\mathbf{X}_t=\mathbf{x}_t}(\mathbf{y})p_{\mathbf{X}_1|\mathbf{X}_t=\mathbf{x}_t}(\mathbf{x}_1)}{p_{\mathbf{Y}|\mathbf{X}_t=\mathbf{x}_t}(\mathbf{y})} = \frac{p_{\mathbf{Y}|\mathbf{X}_1=\mathbf{x}_1}(\mathbf{y})p_{\mathbf{X}_1|\mathbf{X}_t=\mathbf{x}_t}(\mathbf{x}_1)}{p_{\mathbf{Y}|\mathbf{X}_t=\mathbf{x}_t}(\mathbf{y})},$$
(29)

where we used the conditional independence, given $\mathbf{X}_1$, of $\mathbf{X}_t$ and the measurement $\mathbf{Y}$. We assumed that $p_{\mathbf{X}_1|\mathbf{X}_t=\mathbf{x}_t}$ is approximated with $\tilde{p}_{\mathbf{X}_1|\mathbf{X}_t=\mathbf{x}_t} = \mathcal{N}(\hat{\mathbf{x}}_1(\mathbf{x}_t), \nu_t^2 \mathbf{I}_d)$, and we have $p_{\mathbf{Y}|\mathbf{X}_1=\mathbf{x}_1} = \mathcal{N}(\mathbf{H}\mathbf{x}_1, \sigma_n^2 \mathbf{I})$ by construction. Put together, we obtain the approximate density

$$\tilde{p}_{\mathbf{X}_1|\mathbf{X}_t=\mathbf{x}_t,\mathbf{Y}=\mathbf{y}}(\mathbf{x}_1) = \frac{p_{\mathbf{Y}|\mathbf{X}_1=\mathbf{x}_1}(\mathbf{y})\tilde{p}_{\mathbf{X}_1|\mathbf{X}_t=\mathbf{x}_t}(\mathbf{x}_1)}{p_{\mathbf{Y}|\mathbf{X}_t=\mathbf{x}_t}(\mathbf{y})}. \quad (30)$$

Taking the logarithm of equation 30, $-2\log\left(\tilde{p}_{\mathbf{X}_1|\mathbf{X}_t=\mathbf{x}_t,\mathbf{Y}=\mathbf{y}}(\mathbf{x}_1)\right)$ is given by

$$(\mathbf{y} - \mathbf{H}\mathbf{x}_1)^\top \sigma_n^{-2}(\mathbf{y} - \mathbf{H}\mathbf{x}_1) + (\mathbf{x}_1 - \hat{\mathbf{x}}_1(\mathbf{x}_t))^\top \nu_t^{-2}(\mathbf{x}_1 - \hat{\mathbf{x}}_1(\mathbf{x}_t)) + C \quad (31)$$

$$= -2\mathbf{x}_1^\top \left(\nu_t^{-2}\hat{\mathbf{x}}_1(\mathbf{x}_t) + \sigma_n^{-2}\mathbf{H}^\top\mathbf{y}\right) + \mathbf{x}_1^\top \left(\nu_t^{-2}\mathbf{I} + \sigma_n^{-2}\mathbf{H}^\top\mathbf{H}\right)\mathbf{x}_1 + C' \quad (32)$$

$$= -2\mathbf{x}_1^\top \left(\nu_t^{-2}\hat{\mathbf{x}}_1(\mathbf{x}_t) + \sigma_n^{-2}\mathbf{H}^\top\mathbf{y}\right) + \mathbf{x}_1^\top \boldsymbol{\Sigma}_t^{-1}\mathbf{x}_1 + C', \quad (33)$$

where $C, C'$ are independent of $\mathbf{x}_1$ and considered constants and $\boldsymbol{\Sigma}_t = \left(\nu_t^{-2}\mathbf{I}_d + \sigma_n^{-2}\mathbf{H}^\top\mathbf{H}\right)^{-1}$, which is well-defined because $\nu_t^{-2}\mathbf{I}_d + \sigma_n^{-2}\mathbf{H}^\top\mathbf{H}$ is positive-definite. Completing the square with a term independent of $\mathbf{x}_1$, we get

$$-2\log\left(\tilde{p}_{\mathbf{X}_1|\mathbf{X}_t=\mathbf{x}_t,\mathbf{Y}=\mathbf{y}}(\mathbf{x}_1)\right) = (\mathbf{x}_1 - \boldsymbol{\mu}_t)^\top \boldsymbol{\Sigma}_t^{-1}(\mathbf{x}_1 - \boldsymbol{\mu}_t) + C'', \quad (34)$$

where $\boldsymbol{\mu}_t = \boldsymbol{\Sigma}_t \left(\nu_t^{-2}\hat{\mathbf{x}}_1(\mathbf{x}_t) + \sigma_n^{-2}\mathbf{H}^\top\mathbf{y}\right)$ and $C''$ is again a constant independent of $\mathbf{x}_1$. This yields

$$\tilde{p}_{\mathbf{X}_1|\mathbf{X}_t=\mathbf{x}_t,\mathbf{Y}=\mathbf{y}}(\mathbf{x}_1) = e^{-C''/2}\exp\left(-\frac{1}{2}(\mathbf{x}_1 - \boldsymbol{\mu}_t)^\top \boldsymbol{\Sigma}_t^{-1}(\mathbf{x}_1 - \boldsymbol{\mu}_t)\right). \quad (35)$$

As $\tilde{p}_{\mathbf{X}_1|\mathbf{X}_t=\mathbf{x}_t,\mathbf{Y}=\mathbf{y}}$ is a probability density function, $e^{-C''/2}$ corresponds to its normalization factor, which therefore proves that $\tilde{p}_{\mathbf{X}_1|\mathbf{X}_t=\mathbf{x}_t,\mathbf{Y}=\mathbf{y}}(\mathbf{x}_1) = \mathcal{N}(\mathbf{x}_1; \boldsymbol{\mu}_t, \boldsymbol{\Sigma}_t)$. $\qquad\square$

### 8.2.4 PROOF OF PROPOSITION 3

*Proof.* By using the marginal distributions, the general chain rule for joint probability, and independence, we obtain the expression of $p_{\mathbf{X}_{t+\Delta t}|\mathbf{X}_t=\mathbf{x}_t,\mathbf{Y}=\mathbf{y}}(\mathbf{x}_{t+\Delta t})$ as

$$\int_{\mathbb{R}^d} p_{\mathbf{X}_1|\mathbf{X}_t=\mathbf{x}_t,\mathbf{Y}=\mathbf{y}}(\mathbf{x}_1)p_{\mathbf{X}_{t+\Delta t}|\mathbf{X}_1=\mathbf{x}_1,\mathbf{X}_t=\mathbf{x}_t,\mathbf{Y}=\mathbf{y}}(\mathbf{x}_{t+\Delta t}) \, \mathrm{d}\mathbf{x}_1 \quad (36)$$

$$= \int_{\mathbb{R}^d} p_{\mathbf{X}_1|\mathbf{X}_t=\mathbf{x}_t,\mathbf{Y}=\mathbf{y}}(\mathbf{x}_1)p_{\mathbf{X}_{t+\Delta t}|\mathbf{X}_1=\mathbf{x}_1,\mathbf{X}_t=\mathbf{x}_t}(\mathbf{x}_{t+\Delta t}) \, \mathrm{d}\mathbf{x}_1 \quad (37)$$

$$= \int_{\mathbb{R}^d} p_{\mathbf{X}_1|\mathbf{X}_t=\mathbf{x}_t,\mathbf{Y}=\mathbf{y}}(\mathbf{x}_1)\mathcal{N}\left(\mathbf{x}_{t+\Delta t}; (t+\Delta t)\mathbf{x}_1, (1-t-\Delta t)^2\mathbf{I}_d\right) \, \mathrm{d}\mathbf{x}_1, \quad (38)$$

where we used the fact that, conditioned on $\mathbf{X}_1 = \mathbf{x}_1$, $\mathbf{X}_{t+\Delta t} = (1-t-\Delta t)\mathbf{X}_0 + (t+\Delta t)\mathbf{x}_1 \sim \mathcal{N}\left((t+\Delta t)\mathbf{x}_1, (1-t-\Delta t)^2\mathbf{I}_d\right)$. By inserting the expression of the approximation $\tilde{p}_{\mathbf{X}_1|\mathbf{X}_t=\mathbf{x}_t,\mathbf{Y}=\mathbf{y}}(\mathbf{x}_1)$, the approximate density $\tilde{p}_{\mathbf{X}_{t+\Delta t}|\mathbf{X}_t=\mathbf{x}_t,\mathbf{Y}=\mathbf{y}}(\mathbf{x}_{t+\Delta t})$ is given by

$$\int_{\mathbb{R}^d} \mathcal{N}(\mathbf{x}_1; \boldsymbol{\mu}_t, \boldsymbol{\Sigma}_t)\mathcal{N}\left(\mathbf{x}_{t+\Delta t}; (t+\Delta t)\mathbf{x}_1, (1-t-\Delta t)^2\mathbf{I}_d\right) \, \mathrm{d}\mathbf{x}_1 \quad (39)$$

$$= \int_{\mathbb{R}^d} \mathcal{N}(\mathbf{x}_1; \boldsymbol{\mu}_t, \boldsymbol{\Sigma}_t)\mathcal{N}\left(\mathbf{x}_{t+\Delta t} - (t+\Delta t)\mathbf{x}_1; \mathbf{0}, (1-t-\Delta t)^2\mathbf{I}_d\right) \, \mathrm{d}\mathbf{x}_1. \quad (40)$$

This integral can be rewritten as a convolution of two Gaussian distributions, which also yields a Gaussian distribution. Explicitly, we have that

$$\int_{\mathbb{R}^d} \mathcal{N}\left(\mathbf{x}_1; \boldsymbol{\mu}_t, \boldsymbol{\Sigma}_t\right) \mathcal{N}\left(\mathbf{x}_{t+\Delta t} - (t+\Delta t)\mathbf{x}_1; \mathbf{0}, (1-t-\Delta t)^2 \mathbf{I}_d\right) \mathrm{d}\mathbf{x}_1 \tag{41}$$

$$= \int_{\mathbb{R}^d} \mathcal{N}\left(\frac{\mathbf{z}}{t+\Delta t}; \boldsymbol{\mu}_t, \boldsymbol{\Sigma}_t\right) \mathcal{N}\left(\mathbf{x}_{t+\Delta t} - \mathbf{z}; \mathbf{0}, (1-t-\Delta t)^2 \mathbf{I}_d\right) \frac{\mathrm{d}\mathbf{z}}{(t+\Delta t)^d} \tag{42}$$

$$= \int_{\mathbb{R}^d} \mathcal{N}\left(\mathbf{z}; (t+\Delta t)\boldsymbol{\mu}_t, (t+\Delta t)^2 \boldsymbol{\Sigma}_t\right) \mathcal{N}\left(\mathbf{x}_{t+\Delta t} - \mathbf{z}; \mathbf{0}, (1-t-\Delta t)^2 \mathbf{I}_d\right) \mathrm{d}\mathbf{z}. \tag{43}$$

Using the Gaussian convolution identity

$$\int_{\mathbb{R}^d} \mathcal{N}(\mathbf{z}; \boldsymbol{\mu}_1, \boldsymbol{\Sigma}_1) \mathcal{N}(\mathbf{x} - \mathbf{z}; \boldsymbol{\mu}_2, \boldsymbol{\Sigma}_2) \,\mathrm{d}\mathbf{z} = \mathcal{N}(\mathbf{x}; \boldsymbol{\mu}_1 + \boldsymbol{\mu}_2, \boldsymbol{\Sigma}_1 + \boldsymbol{\Sigma}_2), \tag{44}$$

the result simplifies to

$$\tilde{p}_{\mathbf{X}_{t+\Delta t}|\mathbf{X}_t = \mathbf{x}_t, \mathbf{Y} = \mathbf{y}}(\mathbf{x}_{t+\Delta t}) = \mathcal{N}\left(\mathbf{x}_{t+\Delta t}; (t+\Delta t)\boldsymbol{\mu}_t, (t+\Delta t)^2 \boldsymbol{\Sigma}_t + (1-t-\Delta t)^2 \mathbf{I}_d\right), \tag{45}$$

which completes the proof. $\qquad\square$

### 8.2.5 Sampling from the Non-Isotropic Gaussian

We want to show that

$$\boldsymbol{\kappa}_t = \boldsymbol{\Sigma}_t \left(\nu_t^{-1}\boldsymbol{\epsilon}_1 + \sigma_n^{-1}\mathbf{H}^\top \boldsymbol{\epsilon}_2\right) \tag{46}$$

has distribution $\mathcal{N}(0, \boldsymbol{\Sigma}_t)$, where $\boldsymbol{\epsilon}_1 \sim \mathcal{N}(0, \mathbf{I}_d)$ and $\boldsymbol{\epsilon}_2 \sim \mathcal{N}(0, \mathbf{I}_M)$ are independent, and

$$\boldsymbol{\Sigma}_t = \left(\nu_t^{-2}\mathbf{I}_d + \sigma_n^{-2}\mathbf{H}^\top \mathbf{H}\right)^{-1}. \tag{47}$$

Observe that $\boldsymbol{\kappa}_t$ is a Gaussian random vector because it is a linear transform of the independent Gaussians $\boldsymbol{\epsilon}_1$ and $\boldsymbol{\epsilon}_2$. Moreover, since $\boldsymbol{\epsilon}_1$ and $\boldsymbol{\epsilon}_2$ are zero-mean, we have that $\mathbb{E}[\boldsymbol{\kappa}_t] = \mathbf{0}$. Next, we compute the covariance matrix of $\boldsymbol{\kappa}_t$ as

$$\mathrm{Cov}(\boldsymbol{\kappa}_t) = \boldsymbol{\Sigma}_t \,\mathrm{Cov}\left(\nu_t^{-1}\boldsymbol{\epsilon}_1 + \sigma_n^{-1}\mathbf{H}^\top \boldsymbol{\epsilon}_2\right) \boldsymbol{\Sigma}_t. \tag{48}$$

By independence of $\boldsymbol{\epsilon}_1$ and $\boldsymbol{\epsilon}_2$, we obtain that

$$\mathrm{Cov}\left(\nu_t^{-1}\boldsymbol{\epsilon}_1 + \sigma_n^{-1}\mathbf{H}^\top \boldsymbol{\epsilon}_2\right) = \nu_t^{-2}\mathbf{I}_d + \sigma_n^{-2}\mathbf{H}^\top \mathbf{H}, \tag{49}$$

which implies that

$$\mathrm{Cov}(\boldsymbol{\kappa}_t) = \boldsymbol{\Sigma}_t \left(\nu_t^{-2}\mathbf{I}_d + \sigma_n^{-2}\mathbf{H}^\top \mathbf{H}\right) \boldsymbol{\Sigma}_t. \tag{50}$$

But, by definition, $\boldsymbol{\Sigma}_t = \left(\nu_t^{-2}\mathbf{I}_d + \sigma_n^{-2}\mathbf{H}^\top \mathbf{H}\right)^{-1}$, which leads to the desired result, as

$$\mathrm{Cov}(\boldsymbol{\kappa}_t) = \boldsymbol{\Sigma}_t \left(\nu_t^{-2}\mathbf{I}_d + \sigma_n^{-2}\mathbf{H}^\top \mathbf{H}\right) \boldsymbol{\Sigma}_t = \boldsymbol{\Sigma}_t. \tag{51}$$

### 8.3 Extension to More General Noise Types

Throughout this manuscript, isotropic Gaussian noise $\mathbf{n} \sim \mathcal{N}(0, \sigma_n^2 \mathbf{I})$ was considered. Our framework remains valid in the more general setting $\mathbf{n} \sim \mathcal{N}(0, \mathbf{R}_n)$, where $\mathbf{R}_n$ is a symmetric positive definite $M \times M$ covariance matrix, up to the modification of some formulas detailed below.

The results of Proposition 2 become

$$\boldsymbol{\mu}_t(\mathbf{x}_t, \mathbf{y}) = \left(\nu_t^{-2}\mathbf{I}_d + \mathbf{H}^\top \mathbf{R}_n^{-1}\mathbf{H}\right)^{-1} \left(\nu_t^{-2}\hat{\mathbf{x}}_1(\mathbf{x}_t) + \mathbf{H}^\top \mathbf{R}_n^{-1}\mathbf{y}\right), \tag{52}$$

$$\boldsymbol{\Sigma}_t = \left(\nu_t^{-2}\mathbf{I}_d + \mathbf{H}^\top \mathbf{R}_n^{-1}\mathbf{H}\right)^{-1}, \tag{53}$$

since the measurement operation with general noise type implies that $p_{\mathbf{Y}|\mathbf{X}_1=\mathbf{x}_1} = \mathcal{N}(\mathbf{H}\mathbf{x}_1, \mathbf{R}_n)$.

Next, sampling from $\mathcal{N}(0, \boldsymbol{\Sigma}_t)$ is achieved with a method similar to the one of Appendix 8.2.5, with the updated formula

$$\boldsymbol{\kappa}_t = \boldsymbol{\Sigma}_t \left(\nu_t^{-1}\boldsymbol{\epsilon}_1 + \mathbf{H}^\top \mathbf{R}_n^{-\frac{1}{2}}\boldsymbol{\epsilon}_2\right). \tag{54}$$

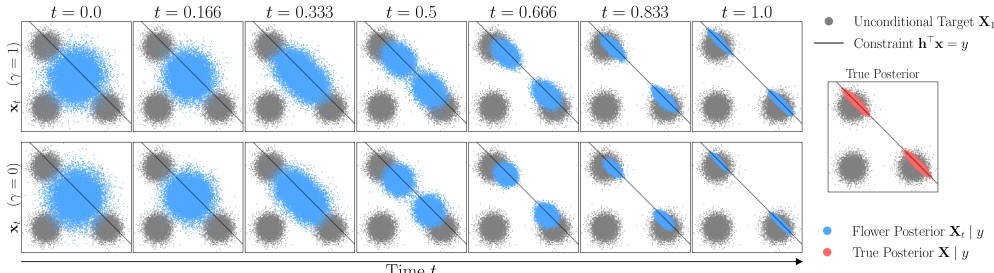

Figure 4: Temporal evolution of 2D *Flower* and comparison with true posterior for noise variance $\sigma_n = 0.25$.

This formula uses the standard definition of the square root of a symmetric positive-definite matrix $\mathbf{A}$. If $\mathbf{A} = \mathbf{PDP}^\top$ is its eigenvalue decomposition, then $\mathbf{A}^{\frac{1}{2}} = \mathbf{PD}^{\frac{1}{2}}\mathbf{P}^\top$, where $\mathbf{D}^{\frac{1}{2}}$ is the diagonal matrix whose entries are the square roots of the nonnegative eigenvalues of $\mathbf{A}$.

As a result, in the measurement-aware destination refinement step of *Flower* described in Section 3, equation 12 now computes $\mathrm{prox}_{\nu_t^2 F_\mathbf{y}}$ with

$$F_\mathbf{y}(\mathbf{x}) = \frac{1}{2}(\mathbf{y} - \mathbf{Hx})^\top \mathbf{R}_n^{-1}(\mathbf{y} - \mathbf{Hx}). \tag{55}$$

In the same step, the matrix $\boldsymbol{\Sigma}_t$ is redefined to be as in equation 53.

In most imaging models, the noise is effectively isotropic. Moreover, if we assume that $\mathbf{R}_n$ is diagonal, the resulting expressions simplify and become straightforward to compute.

## 8.4    NUMERICAL RESULTS EXTENSION

### 8.4.1    TOY EXPERIMENT

The goal of this experiment is to validate our proposed sampling perspective in a setting where samples from the true posterior are available. To this end, we consider a Gaussian mixture model (GMM) as the target distribution of the data $\mathbf{X} \in \mathbb{R}^2$. The forward measurement model consists of a single measurement vector $\mathbf{h} \in \mathbb{R}^d$ (i.e., the forward operator is $\mathbf{H} = \mathbf{h}^\top$) corrupted by additive white Gaussian noise $n \sim \mathcal{N}(0, \sigma_n^2)$, which results in $y = \mathbf{h}^\top \mathbf{x} + n$. In this setup, the posterior distribution $p_{\mathbf{X}|\mathbf{Y}=\mathbf{y}}$ is itself a GMM with known parameters, whose analytical expression is given in equation 57. Geometrically, the noiseless measurement $y = \mathbf{h}^\top \mathbf{x}$ defines a line in the two-dimensional plane with normal vector $\mathbf{h}$. Consequently, when sampling from the posterior $p_{\mathbf{X}|\mathbf{Y}=\mathbf{y}}$, we expect the samples to concentrate on the portions of this line that intersect regions where the prior distribution $p_\mathbf{X}$ has high density.

We trained the unconditional flow-matching vector field using the source $p_{\mathbf{X}_0} = \mathcal{N}(\mathbf{0}, \mathbf{I})$ and target $p_{\mathbf{X}_1} = p_\mathbf{X}$, with further details provided below. We ran *Flower* with $N = 1000$ iterations, which corresponds to the step size $\Delta t = 0.001$. The results are shown in Figure 4, where we used $\mathbf{h}^\top = [1.5, 1.5]$, $\sigma_n = 0.25$, and the observation $y = 1$, and where we report the solution paths of *Flower* with $\gamma \in \{0, 1\}$ alongside true posterior samples. We observe, that when $\gamma = 1$ (the setting required by our theory), *Flower* successfully recovers the samples at $t = 1$, which closely resemble the true posterior. When $\gamma = 0$ (a configuration in which the uncertainty in the destination estimation step is ignored), *Flower* fails to capture samples from the tails of the distribution. In Figure 5, we present another example of *Flower* posterior sampling with $\mathbf{h}^\top = [1.5, -1.5]$, $\sigma_n = 0.75$, and the observation $y = 1$. Once again, we observe that *Flower* successfully generates samples that closely match the true posterior for $\gamma = 1$. In contrast, for $\gamma = 0$, samples from the tails of the true posterior are missing. In Figures 6 and 7, the solution path is illustrated across the successive steps of *Flower* for $\gamma = 0$ and $\gamma = 1$, respectively, which allows us to visualize the dynamics of each step directly.

**Target Prior.** The target distribution of our data $\mathbf{X} \in \mathbb{R}^2$ is a GMM with uniform mixtures given explicitly by

$$p_{\mathbf{X}} = \frac{1}{K} \sum_{k=1}^{K} \mathcal{N}(\boldsymbol{\mu}_k, \boldsymbol{\Sigma}) \tag{56}$$

for some $K \in \mathbb{N}$, $\boldsymbol{\mu}_k \in \mathbb{R}^2$, and $\boldsymbol{\Sigma} \in \mathbb{S}_{++}^2$. Specifically, we use $K = 3$ with $\boldsymbol{\mu}_1 = (-0.25, -0.25)$, $\boldsymbol{\mu}_2 = (-0.25, 0.25)$, $\boldsymbol{\mu}_3 = (0.25, -0.25)$, and covariance matrix $\boldsymbol{\Sigma} = 0.25^2 \mathbf{I}_2$.

**Target Posterior.** The advantage of this setup is that the posterior distribution $p_{\mathbf{X}|\mathbf{Y}=\mathbf{y}}$ can be computed exactly using Bayes' rule. It is given by

$$p_{\mathbf{X}|\mathbf{Y}=\mathbf{y}} = \sum_{k=1}^{K} w_k \mathcal{N}(\boldsymbol{\mu}_{k,\text{post}}, \boldsymbol{\Sigma}_{\text{post}}) \tag{57}$$

where, for all $k = 1, \ldots, K$, $w_k \geq 0$ are some weights and

$$\boldsymbol{\mu}_{k,\text{post}} = \left(\boldsymbol{\Sigma}^{-1} + \sigma_n^{-2} \mathbf{h}\mathbf{h}^{\top}\right)^{-1} \left(\sigma_n^{-2} \mathbf{h}y + \boldsymbol{\Sigma}^{-1} \boldsymbol{\mu}_k\right), \tag{58}$$

$$\boldsymbol{\Sigma}_{\text{post}} = \left(\boldsymbol{\Sigma}^{-1} + \sigma_n^{-2} \mathbf{h}\mathbf{h}^{\top}\right)^{-1}. \tag{59}$$

**Training Details.** The underlying unconditional velocity network is a fully connected network that takes as input a 2D vector and a scalar time, concatenated into a 3D input. It consists of two hidden layers of size 256 with SiLU activations, followed by a final linear layer that outputs a 2D vector. For training, we use a batch size of 2048 with 20000 training steps and a learning rate of $10^{-3}$.

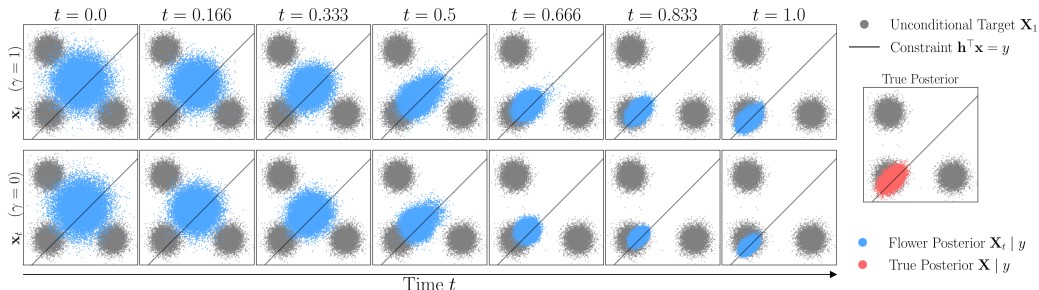

Figure 5: Temporal evolution of 2D *Flower* and comparison with true posterior for noise variance $\sigma_n = 0.75$.

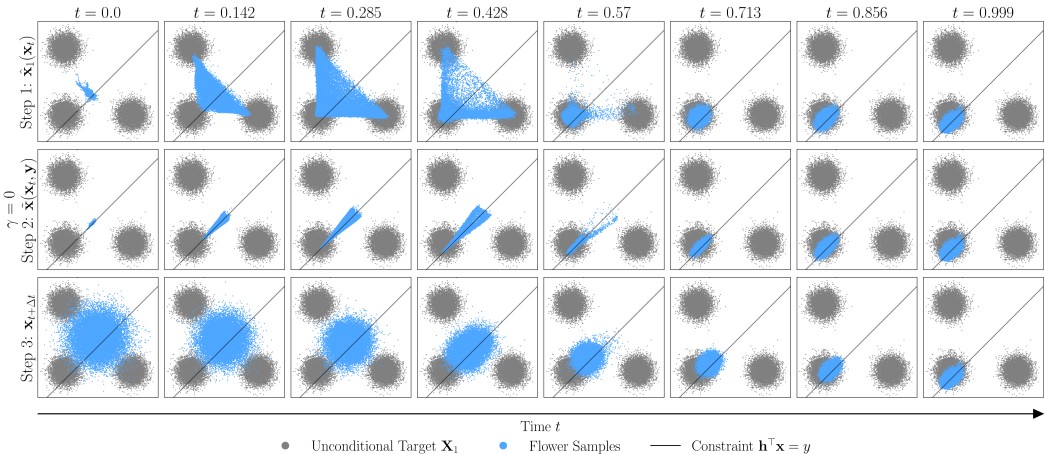

Figure 6: The three steps of 2D *Flower* with temporal evolution for $\gamma = 0$.

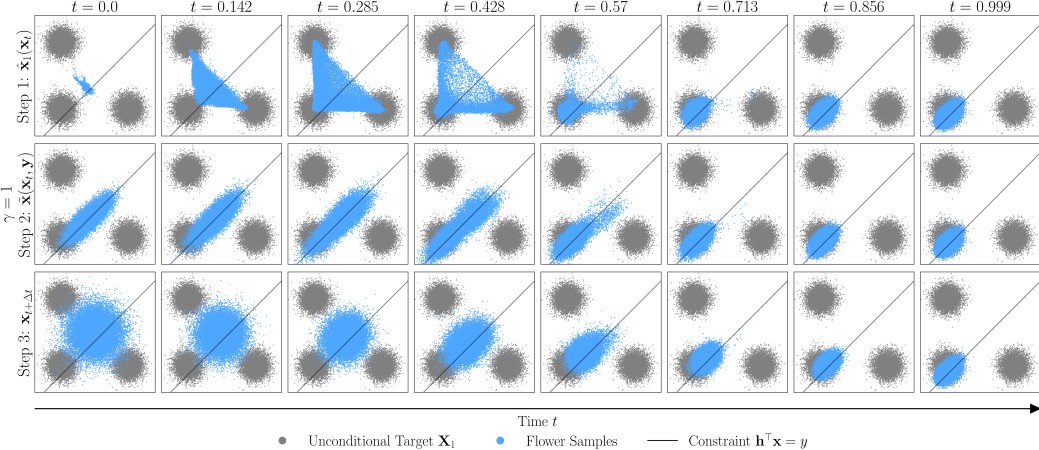

Figure 7: The three steps of 2D *Flower* with temporal evolution for $\gamma = 1$.

## 8.5 BENCHMARK EXPERIMENTS

### 8.5.1 COMPUTATIONAL EFFICIENCY

We report in Table 3 the computational time and memory usage for several methods. Each entry corresponds to the average runtime for the deblurring inverse problem, averaged over 10 CelebA test images of size $128 \times 128$. All experiments were conducted on a Tesla V100-SXM2-32GB GPU.

Table 3: Computation times and memory usage for various methods.

| Method | OT-ODE | D-Flow | Flow Priors | PnP-Flow1 | Flower1 |
|---|---|---|---|---|---|
| Time (s) | 6.549 | 142.18 | 63.771 | 3.020 | 5.622 |
| Memory (GB) | 1.183 | 11.125 | 3.807 | 0.216 | 0.217 |

### 8.5.2 EFFECT OF SOURCE-TARGET COUPLING

Our goal in this section is to benchmark the effect of source–target coupling in the training of the underlying (unconditional) velocity network. To this end, we consider two variants of coupling: one based on mini-batch optimal transport (OT) coupling and the other on independent (IND) coupling. For each variant, we report two settings: a single evaluation of *Flower*; and average over five evaluations. This results in four cases, summarized in Table 4, where we provide results for all inverse problems discussed in the main text, evaluated on 100 test images from the CelebA dataset. For this table, we fix $\gamma = 0$ and $N = 100$.

Table 4: Effect of source-target coupling of the underlying velocity network of *Flower* on different inverse problems on 100 test images of the dataset CelebA.

| Method | Denoising | | | Deblurring | | | Super-resolution | | | Random inpainting | | | Box inpainting | | |
|---|---|---|---|---|---|---|---|---|---|---|---|---|---|---|---|
| | PSNR | SSIM | LPIPS | PSNR | SSIM | LPIPS | PSNR | SSIM | LPIPS | PSNR | SSIM | LPIPS | PSNR | SSIM | LPIPS |
| Degraded | 20.00 | 0.348 | 0.372 | 27.83 | 0.740 | 0.126 | 10.26 | 0.183 | 0.827 | 11.95 | 0.196 | 1.041 | 22.27 | 0.742 | 0.214 |
| Flower1-OT (ours) | 32.28 | 0.914 | 0.034 | 34.98 | 0.947 | 0.026 | 32.36 | 0.923 | 0.034 | 33.08 | 0.944 | 0.018 | 31.19 | 0.945 | 0.022 |
| Flower5-OT (ours) | 33.14 | 0.926 | 0.038 | 35.67 | 0.954 | 0.032 | 33.09 | 0.932 | 0.040 | 33.95 | 0.953 | 0.020 | 31.87 | 0.952 | 0.023 |
| Flower1-IND (ours) | 32.60 | 0.918 | 0.032 | 35.22 | 0.950 | 0.026 | 32.65 | 0.927 | 0.034 | 33.23 | 0.947 | 0.017 | 31.90 | 0.950 | 0.021 |
| Flower5-IND (ours) | 33.48 | 0.930 | 0.037 | 35.90 | 0.957 | 0.031 | 33.41 | 0.935 | 0.039 | 34.24 | 0.955 | 0.020 | 32.78 | 0.958 | 0.022 |

We observe that independent coupling improves the results, which is consistent with our theoretical requirements. Nevertheless, the OT-based variant remains highly competitive, as also illustrated in the main paper. For visual comparison, in Figure 8, we show an example from the deblurring task on an image from the CelebA dataset.

### 8.5.3 EFFECT OF $\gamma$

The hyperparameter $\gamma \in \{0, 1\}$ in *Flower* controls whether the uncertainty of the refinement step (Step 2) is taken into account. While $\gamma = 1$ is required for a Bayesian interpretation of our method, in practice we find that $\gamma = 0$ yields more favorable image-reconstruction metrics. This observation is consistent with the toy experiments, where $\gamma = 0$ led *Flower* to generate samples concentrated in higher-probability regions, while failing to capture the tails of the posterior.

To illustrate this effect, we provide a visual example of the deblurring task on a CelebA image, using the velocity network trained with mini-batch OT coupling for consistency with the numerical results reported in this paper. A single reconstruction with $\gamma = 1$ achieves a PSNR of only $30.84$, compared to $33.01$ for a single reconstruction with $\gamma = 0$ (Figure 8). Moreover, with $\gamma = 1$, it is necessary to average over 100 reconstructions to reach a PSNR comparable to that of $\gamma = 0$, where we average only over 5 reconstructions (Figure 8).

To further illustrate this behavior, we show in Figures 6 and 7 the solution paths of the three *Flower* steps over time for $\gamma = 0$ and $\gamma = 1$, respectively. We observe that $\gamma = 1$ leads to a noisier refinement step. Consequently, we adopt $\gamma = 0$ for all inverse problems, as this provides a more stable refinement step and consistently yields a better quality of reconstruction despite fewer averages.

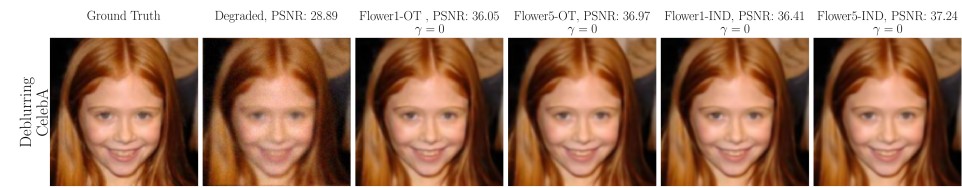

Figure 8: Effect of the source-target coupling for the underlying flow ($\gamma = 0$).

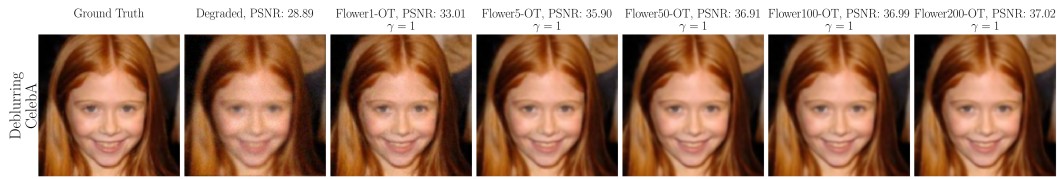

Figure 9: Deblurring results with $\gamma = 1$, obtained by averaging over 1, 5, 50, 100, and 200 runs of *Flower*.

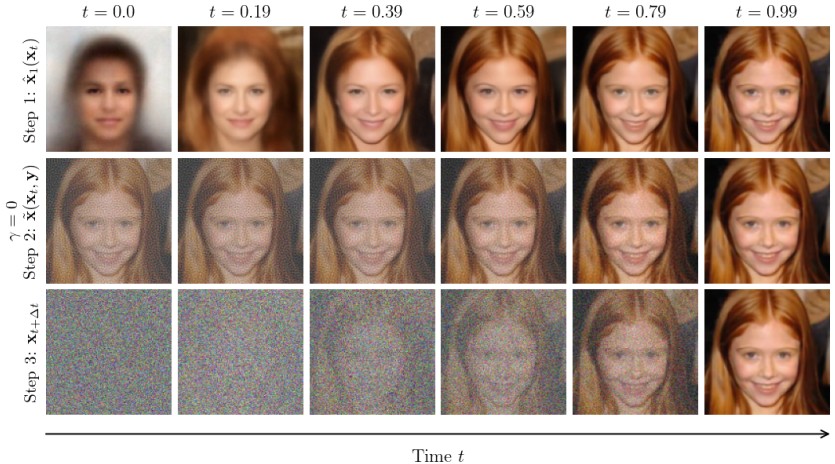

Figure 10: Solution path of *Flower* for deblurring with $\gamma = 0$.

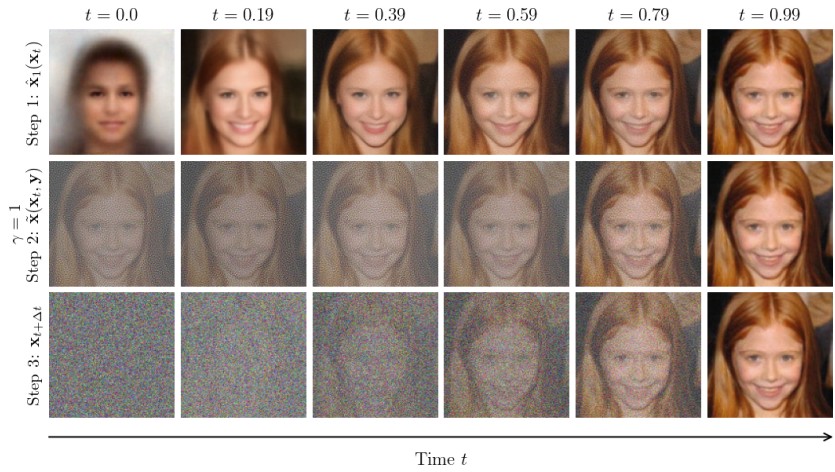

Figure 11: Solution path of *Flower* for deblurring with $\gamma = 1$.

### 8.5.4 EFFECT OF THE NUMBER OF EVALUATIONS FOR AVERAGING

In Tables 1 and 2 of the main paper, we reported results for Flower-1 and Flower-5, which correspond to using a single evaluation and the average over five evaluations of *Flower*, respectively. Here, we ablate the effect of this averaging on both the reconstruction metrics and the computational cost. We focus on the deblurring and random inpainting tasks described in Section 5.1 for the CelebA dataset. Specifically, we vary the number of *Flower* averaging $N_{\text{Avg}}$ from 1 to 10 and report the average PSNR, SSIM, LPIPS, and runtime (in seconds) over 100 CelebA images, measured on a Tesla V100-SXM2-32GB GPU. For the hyperparameters of *Flower*, we use $\gamma = 0$, $N = 100$.

As shown in Figure 12, PSNR and SSIM improve with a steeper gain for smaller numbers of averagings (up to around five) and tend to saturate afterward. LPIPS also increases with averaging, which is undesirable since higher LPIPS indicates worse perceptual quality. The runtime scales linearly with the number of averagings, as expected. Finally, in Figure 13, we present a visual comparison that confirms these trends: averaged reconstructions (e.g., using 5 or 10 evaluations) appear smoother than the single-evaluation result, while the difference between averaging over 5 and 10 evaluations is marginal.

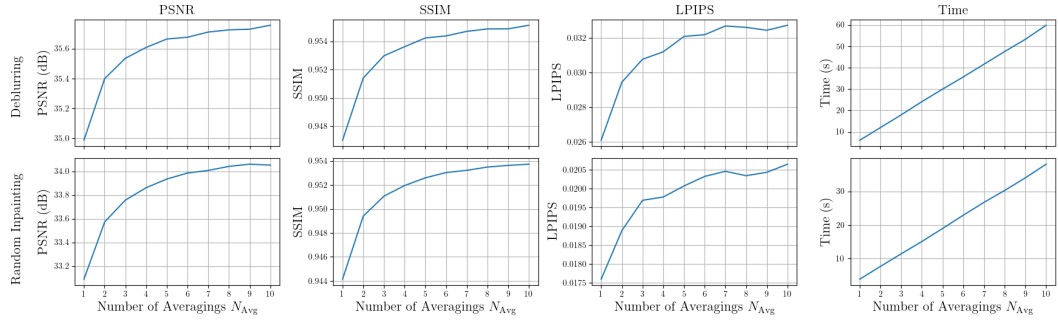

Figure 12: Effect of averaging over different numbers of evaluations of *Flower* on quantitative metrics and computational time.

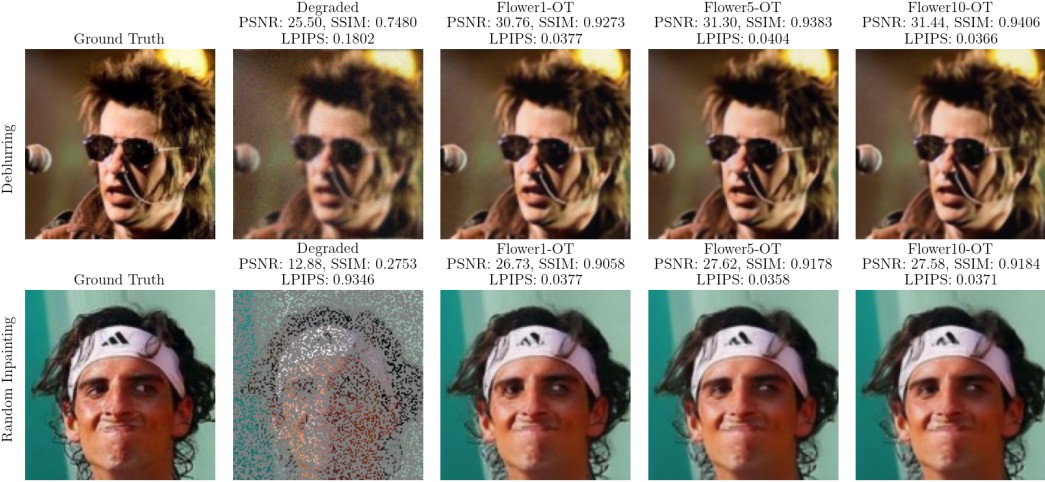

Figure 13: Visual comparison of the effect of averaging over different numbers of evaluations of *Flower*.

### 8.5.5 EFFECT OF ADAPTIVE TIME STEPS

Our theoretical framework does not tie *Flower* to a particular time discretization. This raises the question of whether non-uniform time steps could improve practical performance. To explore this, we compare the uniform (Euler) time discretization with several alternatives. Specifically, we consider a power-law schedule $t_k = \left(\frac{k}{N}\right)^{\alpha}$, where $\alpha = 1$ recovers the uniform grid, $\alpha > 1$ concentrates steps near the start of the trajectory, and $\alpha < 1$ allocates more steps near the end. In our experiments, we use $\alpha = 0.5$ and $\alpha = 2$. We further evaluate a cosine schedule $t_k = \frac{1-\cos(\pi k/N)}{2}$, which yields finer resolution at both the beginning and end of the trajectory. For clarity, Figure 14 visualizes these time grids for $N = 100$. We vary the total number of steps $N \in \{10, 20, 50, 100\}$ and validate these schedules on two restoration tasks (deblurring and random inpainting) shown in Figures 15 and 16. For the hyperparameters of *Flower*, we use $\gamma = 0$, and $N_{\text{Avg}} = 1$. Our results indicate that the $\alpha = 0.5$ schedule achieves noticeably better reconstruction quality with fewer total steps $N$, while $\alpha = 2$ tends to underperform relative to the uniform case. The cosine schedule sometimes provides improvements at low step counts. For larger numbers of steps, all schedules converge to similar performance. Our findings highlight the potential of adaptive time discretizations (with more resolution toward the end of trajectory) to improve quality–compute trade-offs.

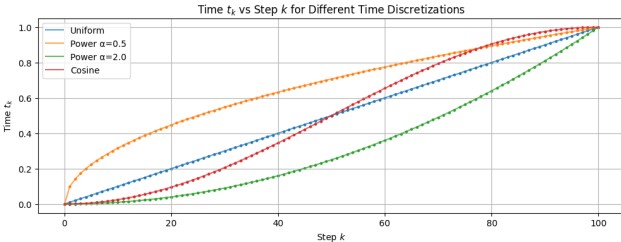

Figure 14: Time $t_k$ vs step $k$ for different time discretizations.

### 8.5.6 INCREASING THE SIZE OF TEST SETS

In the main paper, we evaluated all tasks for each method using 100 CelebA images and 100 AFHQ-Cat images. We chose a test size of 100 because methods such as Flow Priors and D-Flow require backpropagation during inference and are therefore slow. In this appendix, we increase the test set to 1000 images for CelebA and 400 images for AFHQ-Cat in order to further validate *Flower* compared to existing (efficient) flow-matching methods. The results, presented in Tables 5 and 6, show the same trend as in the main paper as *Flower* achieves competitive reconstruction quality.

Table 5: Results on 1000 test images of the dataset CelebA.

| Method | Denoising | | | Deblurring | | | Super-resolution | | | Random inpainting | | | Box inpainting | | |
|---|---|---|---|---|---|---|---|---|---|---|---|---|---|---|---|
| | PSNR | SSIM | LPIPS | PSNR | SSIM | LPIPS | PSNR | SSIM | LPIPS | PSNR | SSIM | LPIPS | PSNR | SSIM | LPIPS |
| Degraded | 20.00 | 0.351 | 0.368 | 27.83 | 0.741 | 0.123 | 10.38 | 0.185 | 0.828 | 12.10 | 0.197 | 1.038 | 22.29 | 0.745 | 0.211 |
| OT-ODE | 30.53 | 0.857 | **0.032** | 33.06 | 0.920 | 0.029 | 31.54 | 0.905 | **0.024** | 28.74 | 0.870 | 0.052 | 29.65 | 0.921 | 0.035 |
| PnP-Flow1 | 31.84 | 0.904 | 0.044 | 34.56 | 0.935 | 0.038 | 31.16 | 0.901 | 0.044 | 33.16 | **0.944** | 0.019 | 30.47 | 0.935 | 0.036 |
| Flower1-OT (ours) | **32.31** | **0.913** | 0.033 | **35.03** | **0.946** | **0.026** | **32.45** | **0.922** | 0.034 | **33.19** | **0.944** | **0.017** | **31.22** | **0.946** | **0.021** |

Table 6: Results on 400 test images of the dataset AFHQ-Cat.

| Method | Denoising | | | Deblurring | | | Super-resolution | | | Random inpainting | | | Box inpainting | | |
|---|---|---|---|---|---|---|---|---|---|---|---|---|---|---|---|
| | PSNR | SSIM | LPIPS | PSNR | SSIM | LPIPS | PSNR | SSIM | LPIPS | PSNR | SSIM | LPIPS | PSNR | SSIM | LPIPS |
| Degraded | 20.00 | 0.315 | 0.517 | 24.03 | 0.516 | 0.452 | 11.88 | 0.217 | 0.881 | 13.55 | 0.231 | 1.071 | 21.80 | 0.741 | 0.203 |
| OT-ODE | 30.01 | 0.814 | **0.077** | 27.06 | 0.711 | **0.126** | 25.92 | 0.715 | **0.109** | 29.38 | 0.839 | 0.091 | 24.77 | 0.875 | 0.085 |
| PnP-Flow1 | 31.17 | 0.862 | 0.136 | 27.94 | 0.759 | 0.306 | **26.96** | **0.762** | 0.170 | **33.01** | **0.918** | **0.037** | 26.46 | 0.897 | 0.102 |
| Flower1-OT (ours) | **31.66** | **0.878** | 0.104 | **28.63** | **0.773** | 0.255 | 26.24 | 0.740 | 0.273 | 32.98 | **0.918** | 0.040 | **26.68** | **0.915** | **0.062** |

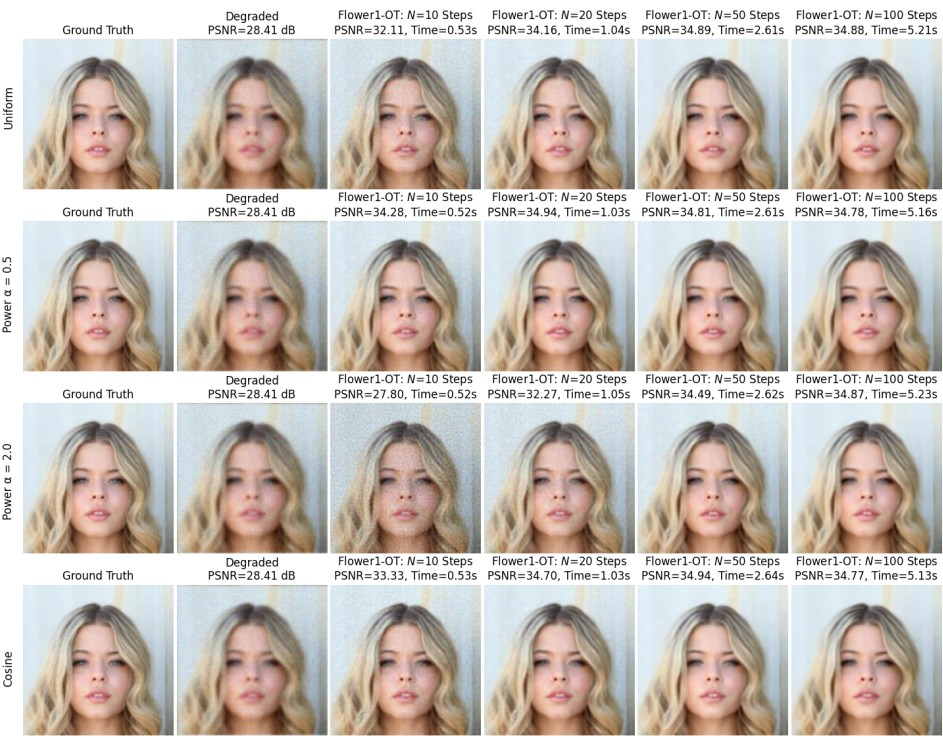

Figure 15: Deblurring example for uniform versus adaptive solvers.

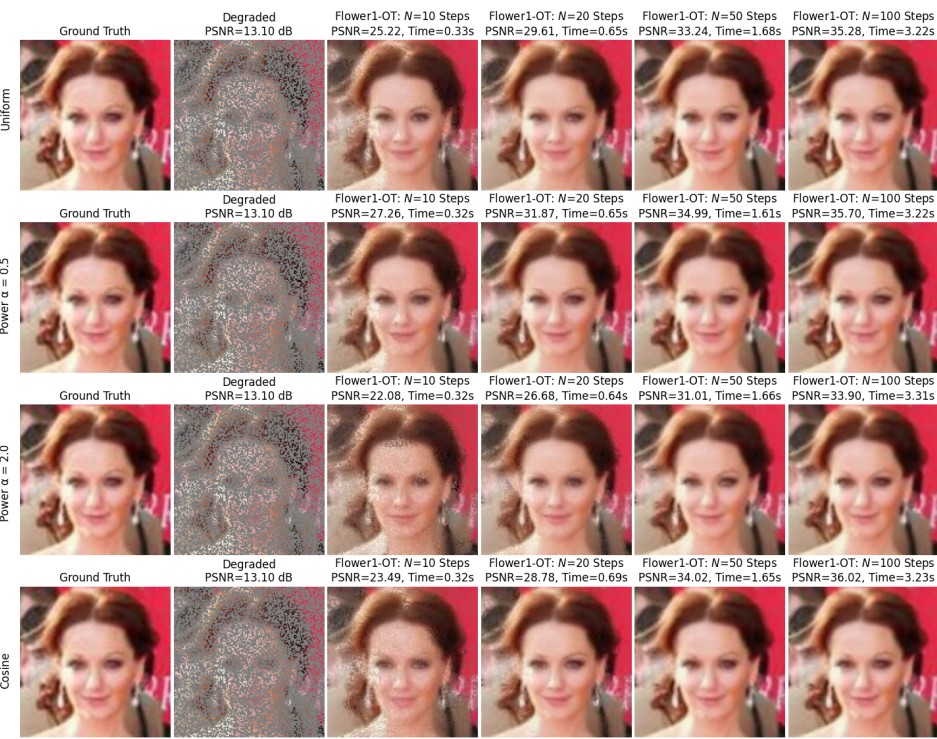

Figure 16: Random inpainting example for uniform versus adaptive solvers.

### 8.5.7 APPLICATION TO FURTHER INVERSE PROBLEMS

We aim to validate the generality of our framework by evaluating it on a wider range of inverse problems.

**Compressed Sensing Fourier Sampling.** We choose a forward operator inspired by magnetic resonance imaging (MRI) reconstruction: the Fourier transform followed by a binary sampling mask. We use two masks: (i) a Cartesian mask with a sampling ratio of $0.2188$, and (ii) a radial mask with a sampling ratio of $0.2990$. The measurements are corrupted with additive white Gaussian noise of standard deviation $\sigma_n = 0.002$. For our examples, we use two images from the AFHQ-Cat dataset. For the hyperparameters of *Flower*, we use $\gamma = 0$, $N = 100$, and $N_{\text{Avg}} = 1$. Since the images are RGB, we apply the forward operator channel by channel. We summarize our results in Figure 17. In both cases, we observe that *Flower* successfully handles this inverse problem and produces reconstructions that improve on the zero-filled baseline.

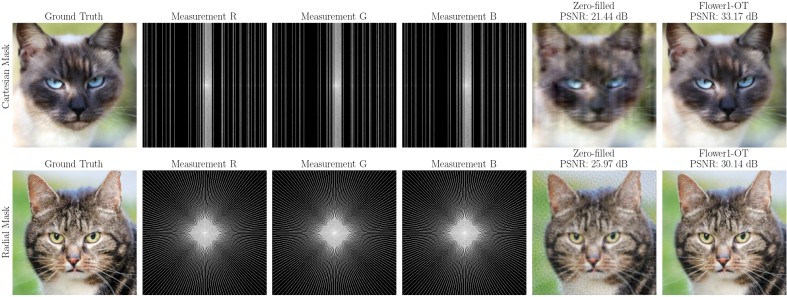

Figure 17: Reconstruction results for compressed sensing with Cartesian and radial masks.

**Non-Isotropic Gaussian Noise.** Here, we aim to verify the theoretical results from Appendix 8.3 for non-isotropic additive Gaussian noise using numerical experiments. For demonstration, we focus on two tasks: image denoising and random inpainting using the same setup described in Section 5 for the CelebA images, except for the noise dynamics. Here, we add non-isotropic Gaussian noise to the image by applying Gaussian noise with $\sigma_n = 3$ to the central box of size $(64 \times 64)$ of the $(128 \times 128)$ image and $\sigma_n = 1$ outside this box. For the *Flower* hyperparameters, we use $\gamma = 0$, $N = 100$, and $N_{\text{Avg}} = 1$. We observe that, consistent with our theory, we are able to recover good-quality reconstructions given a non-isotropic Gaussian noise.

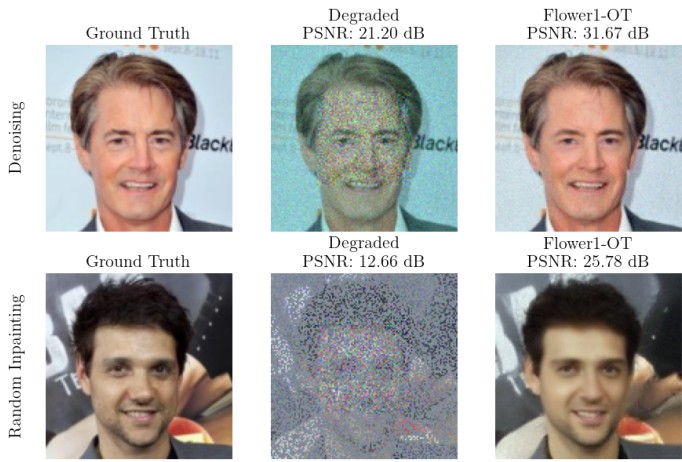

Figure 18: Visual results for inverse problems with non-isotropic Gaussian noise.

### 8.5.8 Hyperparameters for All Methods

In Tables 7 and 8, we report the hyperparameters that we used for all methods. Most of the hyperparameters are adapted from Martin et al. (2025).

Table 7: Hyperparameters for all methods on the CelebA dataset.

| Method | Hyperparameters | Denoising | Deblurring | Super-resolution | Random inpainting | Box inpainting |
|---|---|---|---|---|---|---|
| DiffPIR | $\zeta$ (blending) | 1.0 | 1.0 | 1.0 | 1.0 | N/A |
| | $\lambda$ (regularization) | 1.0 | 1000.0 | 100.0 | 1.0 | N/A |
| PnP-GS | $\gamma$ (learning rate) | - | 2.0 | 2.0 | 1.0 | N/A |
| | $\alpha$ (inertia param.) | 1.0 | 0.5 | 1.0 | 0.5 | N/A |
| | $\sigma_f$ (factor for noise input) | 1.0 | 1.8 | 3.0 | 1.0 | N/A |
| | $n_{\text{iter}}$ (number of iter.) | 1 | 35 | 20 | 23 | N/A |
| OT-ODE | $t_0$ (initial time) | 0.3 | 0.4 | 0.1 | 0.1 | 0.1 |
| | $\gamma$ | time-dependent | time-dependent | constant | constant | time-dependent |
| Flow-Priors | $\lambda$ (regularization) | 100 | 1,000 | 10,000 | 10,000 | 10,000 |
| | $\eta$ (learning rate) | 0.01 | 0.01 | 0.1 | 0.01 | 0.01 |
| D-Flow | $\lambda$ (regularization) | 0.001 | 0.001 | 0.001 | 0.01 | 0.001 |
| | $\alpha$ (blending) | 0.1 | 0.1 | 0.1 | 0.1 | 0.1 |
| | $n_{\text{iter}}$ (number of iter.) | 3 | 7 | 10 | 20 | 9 |
| PnP-Flow1 | $\alpha$ (learning-rate factor) | 0.8 | 0.01 | 0.3 | 0.01 | 0.5 |
| | $N$ (Number of time steps) | 100 | 100 | 100 | 100 | 100 |
| | $N_{\text{Avg}}$ (Number of averagings) | 1 | 1 | 1 | 1 | 1 |
| PnP-Flow5 | $\alpha$ (learning-rate factor) | 0.8 | 0.01 | 0.3 | 0.01 | 0.5 |
| | $N$ (Number of time steps) | 100 | 100 | 100 | 100 | 100 |
| | $N_{\text{Avg}}$ (Number of averagings) | 5 | 5 | 5 | 5 | 5 |
| Flower1-OT | $\gamma$ (refinement uncertainty) | 0 | 0 | 0 | 0 | 0 |
| | $N$ (Number of time steps) | 100 | 100 | 100 | 100 | 100 |
| | $N_{\text{Avg}}$ (Number of averagings) | 1 | 1 | 1 | 1 | 1 |
| Flower5-OT | $\gamma$ (refinement uncertainty) | 0 | 0 | 0 | 0 | 0 |
| | $N$ (Number of time steps) | 100 | 100 | 100 | 100 | 100 |
| | $N_{\text{Avg}}$ (Number of averagings) | 5 | 5 | 5 | 5 | 5 |

Table 8: Hyperparameters for all methods on the AFHQ-Cat dataset.

| Method | | Denoising | Deblurring | Super-resolution | Random inpainting | Box inpainting |
|---|---|---|---|---|---|---|
| DiffPIR | $\zeta$ (blending) | 1.0 | 1.0 | 1.0 | 1.0 | N/A |
| | $\lambda$ (regularization) | 1.0 | 1000.0 | 100.0 | 1.0 | N/A |
| PnP-GS | $\gamma$ (learning rate) | - | 2.0 | 2.0 | 1.0 | N/A |
| | $\alpha$ (inertia param.) | 1.0 | 0.3 | 1.0 | 0.5 | N/A |
| | $\sigma_f$ (factor for noise input) | 1.0 | 1.8 | 5.0 | 1.0 | N/A |
| | $n_{\text{iter}}$ (number of iter.) | 1 | 60 | 50 | 23 | N/A |
| OT-ODE | $t_0$ (initial time) | 0.3 | 0.3 | 0.1 | 0.1 | 0.1 |
| | $\gamma$ | time-dependent | time-dependent | constant | constant | time-dependent |
| Flow-Priors | $\lambda$ (regularization) | 100 | 1,000 | 10,000 | 10,000 | 10,000 |
| | $\eta$ (learning rate) | 0.01 | 0.01 | 0.1 | 0.01 | 0.01 |
| D-Flow | $\lambda$ (regularization) | 0.001 | 0.01 | 0.001 | 0.001 | 0.01 |
| | $\alpha$ (blending) | 0.1 | 0.5 | 0.1 | 0.1 | 0.1 |
| | $n_{\text{iter}}$ (number of iter.) | 3 | 20 | 20 | 20 | 9 |
| PnP-Flow1 | $\alpha$ (learning-rate factor) | 0.8 | 0.01 | 0.01 | 0.01 | 0.5 |
| | $N$ (Number of time steps) | 100 | 500 | 500 | 200 | 100 |
| | $N_{\text{Avg}}$ (Number of averagings) | 1 | 1 | 1 | 1 | 1 |
| PnP-Flow5 | $\alpha$ (learning-rate factor) | 0.8 | 0.01 | 0.01 | 0.01 | 0.5 |
| | $N$ (Number of time steps) | 100 | 500 | 500 | 200 | 100 |
| | $N_{\text{Avg}}$ (Number of averagings) | 5 | 5 | 5 | 5 | 5 |
| Flower1-OT | $\gamma$ (refinement uncertainty) | 0 | 0 | 0 | 0 | 0 |
| | $N$ (Number of time steps) | 100 | 100 | 500 | 200 | 100 |
| | $N_{\text{Avg}}$ (Number of averagings) | 1 | 1 | 1 | 1 | 1 |
| Flower5-OT | $\gamma$ (refinement uncertainty) | 0 | 0 | 0 | 0 | 0 |
| | $N$ (Number of time steps) | 100 | 100 | 500 | 200 | 100 |
| | $N_{\text{Avg}}$ (Number of averagings) | 5 | 5 | 5 | 5 | 5 |

