# OpenReview forum: "Flower: A Flow-Matching Solver for Inverse Problems"
_ICLR.cc/2026/Conference — ICLR 2026 Poster_

### Official Review · Reviewer_Kgmh · 2025-10-27

**Soundness:** 4
**Presentation:** 4
**Contribution:** 4
**Rating:** 8
**Confidence:** 3

**Summary:**

The paper “FLOWER: A Flow-Matching Solver for Inverse Problems” introduces Flower, a general-purpose solver that leverages a pre-trained flow model to reconstruct signals consistent with given measurements. Flower operates through a three-step iterative process: (1) flow-consistent destination estimation, where the velocity network predicts a denoised target; (2) measurement-aware refinement, projecting the estimate onto the feasible set defined by the forward operator; and (3) time progression, updating the trajectory along the flow path. The authors provide a Bayesian interpretation, showing that Flower approximates posterior sampling, thereby linking plug-and-play (PnP) and generative flow-based solvers within a unified framework. Empirically, Flower achieves state-of-the-art reconstruction quality across diverse inverse problems using nearly identical hyperparameters. Its main contribution lies in bridging Bayesian posterior sampling and PnP methods through a theoretically grounded and practically efficient flow-matching formulation.

**Strengths:**

The paper:
-  Introduces a novel three-step flow-matching framework (Flower) for inverse problems, bridging generative flow models and plug-and-play (PnP) methods under a unified Bayesian view.

-  Provides solid theoretical grounding with a clear Bayesian justification and strong empirical results showing state-of-the-art/competitive reconstructions across diverse tasks.

- The paper is well-structured and accessible, clearly explaining algorithmic steps and visualizing the process through intuitive figures.

**Weaknesses:**

-  **More ablation** could be conducted on the number of evaluations performed at each step. The paper currently presents results for Flower1 and Flower5, where Flower5 shows steady improvement over Flower1. An ablation plot illustrating this trend would strengthen the analysis.

-  Although the method claims hyperparameter robustness, the validation is limited to standard imaging tasks. Demonstrating applications to **nonlinear** or **real-world** inverse problems (e.g., MRI, or compressed sensing) would further validate its generality and impact.

**Questions:**

- Could the authors provide an ablation plot showing the effect of varying the number of evaluations per step (e.g., Flower1-10?) to better illustrate the trade-off between reconstruction quality and computational cost?

- The current method is compatible with the standard Euler sampler used in flow matching. Could the authors discuss whether other ODE or higher-order samplers (e.g., Heun or adaptive solvers) could be integrated into Flower, and what potential advantages or challenges might arise?

- Could the authors elaborate on possible extensions of Flower to nonlinear or real-world inverse problems (e.g., MRI reconstruction, phase retrieval, or compressed sensing)? In particular, what modifications or additional components would be required to handle these more complex forward models?

---

> ### Author Response · Authors · 2025-11-20
>
> We thank the reviewer for their time and feedback. We are glad to hear the recognition of our solid Bayesian justification and strong empirical results.
>
> Below, we address each raised questions and summarize the related changes in the revised manuscript.
>
> > [Weakness] More ablation could be conducted on the number of evaluations performed at each step. The paper currently presents results for Flower1 and Flower5, where Flower5 shows steady improvement over Flower1. An ablation plot illustrating this trend would strengthen the analysis.
>
> > [Related Question] Could the authors provide an ablation plot showing the effect of varying the number of evaluations per step (e.g., Flower1-10?) to better illustrate the trade-off between reconstruction quality and computational cost?
>
> (a). We conducted the requested ablation by varying the number of Flower evaluations for averaging from 1 to 10. The new plots are included in Appendix 8.5.4. As shown there, PSNR and SSIM increase steeply for 1–5 evaluations (averages) and begin to saturate afterward. LPIPS, on the other hand, degrades for larger numbers, and the runtime increases linearly. We provide visual comparisons in Figure 13 to further clarify the cost–performance trade-off.
>
>
> > [Question] The current method is compatible with the standard Euler sampler used in flow matching. Could the authors discuss whether other ODE or higher-order samplers (e.g., Heun or adaptive solvers) could be integrated into Flower, and what potential advantages or challenges might arise?
>
> (b). Higher order solvers (e.g., Heun): Step 1 of Flower follows directly from the conditional mean calculation $ \hat{\mathbf{x}}\_1 (\mathbf{x}\_{t}) =\\mathbb{E}[\mathbf{X}\_{1} \\mid \mathbf{X}\_{t} = \mathbf{x}\_{t}]
> = \mathbf{x}\_{t} + (1 - t)\\,\mathbf{v}^{\\theta}(\mathbf{x}\_{t}, t).
> $ From an ODE perspective, this step resembles taking a single Euler step of size $(1-t)$. One could hope that replacing this update with a higher-order ODE solver could improve reconstruction quality with fewer total steps. For instance, we could attempt to imitate a Heun update by replacing
> $
> \hat{\mathbf{x}}\_{1} = \mathbf{x}\_{t} + (1 - t)\\,\mathbf{v}\_{t}(\mathbf{x}\_{t})
> $
> with the two steps (i)
> $
> \tilde{\mathbf{x}}\_{1}
> =  \mathbf{x}\_{t} + (1 - t)\\,\mathbf{v}\_{t}(\mathbf{x}\_{t})
> $
> and, (ii) $\hat{\mathbf{x}}\_{1} = \mathbf{x}\_{t} + 0.5 (1 -t) ( \mathbf{v}\_{t}(\mathbf{x}\_{t})) + \mathbf{v}\_1(\tilde{\mathbf{x}}\_{1} )). $ However, this modification conflicts with the conditional-mean structure that Step 1 is designed to exploit. In particular, the vector field at $t = 1$ carries no meaningful information. The design of a principled higher-order method compatible with the conditional-mean formulation remains an open direction for future research.
>
> Adaptive Solvers: Adaptive solvers build upon non-uniform time discretization. To test whether non-uniform time discretization can improve efficiency, we added new experiments in Appendix 8.5.5. We evaluate several adaptive schedules and find that using coarse steps early and finer steps near $t=1$ improves reconstruction quality for a fixed step budget, reducing runtime. Results are summarized in Figures 15 and 16 of the revised manuscript.
>
> > [Weakness] Although the method claims hyperparameter robustness, the validation is limited to standard imaging tasks. Demonstrating applications to nonlinear or real-world inverse problems (e.g., MRI, or compressed sensing) would further validate its generality and impact.
>
> > [Related Question] Could the authors elaborate on possible extensions of Flower to nonlinear or real-world inverse problems (e.g., MRI reconstruction, phase retrieval, or compressed sensing)? In particular, what modifications or additional components would be required to handle these more complex forward models?
>
> Flower already supports general linear forward operators, including those used in MRI. To illustrate this, we have added a compressed sensing Fourier experiment using masks derived from MRI acquisition patterns. As shown in Appendix 8.5.7 (“Compressed Sensing Fourier Sampling”) of the updated manuscript, Flower achieves clear improvements over a zero-filled reconstruction with the same hyperparameters as other tasks.
>
> (c). Extending Flower to nonlinear inverse problems is more challenging, as Proposition 2 no longer applies and the approximate posterior $ \tilde{p}\_{\mathbf{X}\_1 \\mid \mathbf{X}\_t = \mathbf{x}\_t , \mathbf{Y}= \mathbf{y}}$ is no longer Gaussian. Nevertheless, one could still compute the score of $ \tilde{p}\_{\mathbf{X}\_1 \\mid \mathbf{X}\_t = \mathbf{x}\_t , \mathbf{Y}= \mathbf{y}}$ and apply Langevin dynamics to sample from it. We discuss these considerations in Section 6 of the updated manuscript.

---

> ### Comment · Reviewer_Kgmh · 2025-11-25
>
> Thank you for the insightful and detailed responses. My original concerns have been largely addressed.
>
> I also reviewed the questions raised by the other reviewers, and I believe the authors did an excellent job clarifying the key points, particularly the new 1,000-image experiments, which demonstrate that the proposed method performs robustly across a wide range of settings. I also found the ablation on $N_{avg}$ interesting, as SSIM and PSNR steadily increase, suggesting the MAP estimate quite well. Additionally, the paper’s task coverage is comparable to prior baselines, and it offers both a solid theoretical contribution and a clear, well-structured presentation. Overall, tackling (linear) inverse problems with flow matching is a promising and timely direction, and this work provides a solid foundation for future research and improvement in this area.
>
> In light of these clarifications and additions, I would like to increase my confidence score to enhance the chance of this paper being accepted.

---

### Official Review · Reviewer_rzV5 · 2025-10-31

**Soundness:** 3
**Presentation:** 3
**Contribution:** 3
**Rating:** 6
**Confidence:** 4

**Summary:**

Flower is a plug-and-play solver for linear inverse problems with pre-trained flow matching models. Given a pre-trained velocity model, Flower solves a linear inverse problem by iterating over three steps:
1. Estimation of clean image $x_1$ given the current noisy image and velocity model.
2. Projection of the estimated image $x_1$ into a feasible set with a proximal operator
3. Estimation of the next noisy image along the flow using the projected $x_1$.

The correctness of the proposed solver has been explained through Bayesian analysis and the key result shows that Flower samples from an approximate posterior distribution.

**Strengths:**

1. The core method has been explained intuitively and seems easy to implement.
2. Computational efficiency: Runtime of Flower is less compared to many other baselines as indicated in Table 4.
3. Robustness: The method uses same hyper parameters for all the linear inverse problems included in the paper. This simplifies hyper parameter tuning.

**Weaknesses:**

1. Writing should be more specific.
    1. Contributions should indicate that this is a solver for linear inverse problems. Statements such as “Thus, Flower can be seen as a plug-and-play inverse solver with general application, with our Bayesian justification establishing a novel link between posterior sampling and the PnP frameworks in generative models.” need to be updated to clarify that this is a solver for linear inverse problems.
    2. The paper should also indicate that the Bayesian analysis is only valid for sampling from the approximate posterior and not the true posterior.
2. Only 100 images have been used for quantitative results which is a really small test dataset.

**Questions:**

1. Estimating $\Sigma_t$ involves computing an inverse of a matrix. How do you efficiently solve this such that the solver itself is light weight?
2. Line 342 states that the validation set for AFHQ-Cat was constructed with 32 images. However, the caption of Table 2 suggests that 100 test images were used. One of these statements need to be updated.
3. What is $m_t$ in line 715? Is it $\hat{x}_1(x_t)$? If that is indeed the case, it’s more readable to use $\hat{x}_1(x_t)$ instead.

---

> ### Author Response · Authors · 2025-11-20
>
> We thank the reviewer for their time and feedback. We are pleased to hear the reviewer’s positive remarks regarding the clarity, efficiency, and robustness of our proposed solver.
>
> Below, we address all the raised concerns point by point.
>
>  > [Weakness] Writing should be more specific...
>
> (a). We revised the manuscript and adapted the writing to be more specific about linear inverse problems and the approximate posterior. Please let us know if any further changes are required.  We also discuss a possible extension of Flower to nonlinear inverse problems in Section 6.
>
> > [Weakness] Only 100 images have been used for quantitative results which is a really small test dataset.
>
> (b). The reviewer is correct that our original evaluation used 100 images, largely because several baselines are very slow (e.g., Flow-Priors, D-Flow). To address the concern, we expanded our test sets following the reviewer’s suggestion:
>
> - CelebA: 1000 test images
> - AFHQ-Cat: 400 test images
>
> We report the new results in the revised paper (Section 8.5.6 of the Appendix), and we present them in the rebuttal in response point (a) to reviewer kQup. The performance trends remain consistent with the 100-image evaluation: Flower maintains top or near-top performance across tasks. We also added experiments on applications of Flower to Fourier sampling and non-isotropic Gaussian noise in Appendix 8.5.7.
>
>
> > [Question] Estimating $\Sigma_t$ involves computing an inverse of a matrix. How do you efficiently solve this such that the solver itself is light weight?
>
> (c). We have added the following implementation details for clarity in Section 5 of the revised manuscript. Indeed, computing $\mathbf{\Sigma}_t \mathbf{b}$ for any $\mathbf{b} \in \mathbb{R}^d$ requires evaluating the inverse operation $(\nu_t^{-2} \mathbf{I} + \sigma_n^{-2} \mathbf{H}^{\top} \mathbf{H})^{-1} \mathbf{b}$. Wherever this inversion is needed, we instead solve the corresponding linear system $(\nu_t^{-2} \mathbf{I} + \sigma_n^{-2} \mathbf{H}^{\top} \mathbf{H}) \mathbf{z} = \mathbf{b}$ using conjugate gradients (CG), and we return $\mathbf{z}$ as the result of the operation. The CG implementation is efficient due to the positive-definite structure of $\nu_t^{-2} \mathbf{I} + \sigma_n^{-2} \mathbf{H}^{\top} \mathbf{H}$ and typically converges within a few iterations, remaining cheaper than a single evaluation of the velocity network.
>
> > [Question] Line 342 states that the validation set for AFHQ-Cat was constructed with 32 images. However, the caption of Table 2 suggests that 100 test images were used. One of these statements need to be updated.
>
> (d). We apologize for not being clear enough. We revised Section 5.1 to state this clearly. Specifically,
>
> - 32 images → used only for validation, i.e., for tuning the hyperparameters of all methods.
> - 100 images → used exclusively for testing and reporting metrics, using the same hyperparameters tuned on the validation images.
>
> This separation is crucial for fair comparison with the baseline methods.
>
> > [Question] What is $m_t$ in line 715?
>
> (e). Thank you for catching this typographical error that has been corrected.

---

### Official Review · Reviewer_kQup · 2025-10-31

**Soundness:** 2
**Presentation:** 2
**Contribution:** 2
**Rating:** 4
**Confidence:** 2

**Summary:**

The paper proposes a solver for inverse problems by leveraging pre-trained flow models to produce reconstructions of observed measurements. The proposed solver use pretrained network to predict the target, which is then refined by a proximal step and noise-injection step. The paper provides theoretical analysis showing that the process aproximates bayesian posterior sampling, and provides empirical results on CelebA and AFHQ-cat datasets.

**Strengths:**

+ The paper proposes a new inverse problem solver that unifies PnP intuition with a Bayesian posterior-sampling narrative.
+ Although I didn't check all the math details presented in the paper, it seems to provide a decent theoretical connection to bayesian posterior sampling using \piGDM-style gaussian approximation.
+ The practicality of this method seems good, with runtime comparable to PnP-Flow and OT-ODE; memory light (tab. 4).

**Weaknesses:**

- The empirical contribution of this submission, though, seems relatively weak. The main numerical experiments are conducted on two small scale datasets, 100 samples from CelebA and AFHQ-Cat, respectively.
- The overall improvements upon previous methods seem not very significant (in tab. 1 & 2).
- In terms of the limitation of assumptions, i) the paper focuses only on linear problems, and ii) if I understand correctly the proximal step further assumes isotropic prior and measurement operation, which could be oversimplified.
- The most stand-out distinctions from previous flow-based inverse problem solvers are not very clear to me.
- Admittely, I am not very familiar with the current progress of this particular sub-domain. I would therefore also like to hear other colleague reviewers' opinions.

**Questions:**

Please refer to the weaknesses section for questions if any. Thanks.

---

> ### Author Response · Authors · 2025-11-20
>
> We thank the reviewer for their feedback. We are glad that they appreciate that Flower unifies PnP intuition with a Bayesian perspective and recognize that our approach is practical.
>
> We now address each concern in detail.
>
> > The empirical contribution of this submission, though, seems relatively weak. The main numerical experiments are conducted on two small scale datasets, 100 samples from CelebA and AFHQ-Cat, respectively.
>
> (a). The 100 test image evaluation followed prior work because several baselines are computationally expensive. To address this limitation, we have added new experiments for the faster solvers (Flower, PnP-Flow, and OT-ODE) on
>  - 1000 test images from CelebA, and
>  - 400 test images from AFHQ-Cat.
>
> These results (now included in Appendix 8.5.6) show the same trends as the main paper: Flower matches or surpasses other methods across most of the tasks. For clarity, we also present the new tables below.
>
> Results on 1000 test images of CelebA.
>
> |Method|Denoising|Deblurring|Super-resolution|Random-inpainting|Box-inpainting|
> |-|-|-|-|-|-|
> |Metrics|PSNR SSIM LPIPS|PSNR SSIM LPIPS|PSNR SSIM LPIPS|PSNR SSIM LPIPS|PSNR SSIM LPIPS|
> |Degraded|20.00 0.351 0.368|27.83 0.741 0.123|10.38 0.185 0.828|12.10 0.197 1.038|22.29 0.745 0.211|
> |OT-ODE|30.53 0.857 **0.032**|33.06 0.920 *0.029*|*31.54* *0.905* **0.024**|28.74 *0.870* 0.052|29.65 0.921 *0.035*|
> |PnP-Flow1|*31.84* *0.904* 0.044|*34.56* *0.935* 0.038|31.16 0.901 0.044|*33.16* **0.944** *0.019*|*30.47* *0.935* 0.036|
> |Flower1-OT(ours)|**32.31** **0.913** *0.033*|**35.03** **0.946** **0.026**|**32.45** **0.922** *0.034*|**33.19** **0.944** **0.017**|**31.22** **0.946** **0.021**|
>
> Results on 400 test images of AFHQ-Cat.
>
> |Method|Denoising|Deblurring|Super-resolution|Random-inpainting|Box-inpainting|
> |-|-|-|-|-|-|
> |Metrics|PSNR SSIM LPIPS|PSNR SSIM LPIPS|PSNR SSIM LPIPS|PSNR SSIM LPIPS|PSNR SSIM LPIPS|
> |Degraded|20.00 0.315 0.517|24.03 0.516 0.452|11.88 0.217 0.881|13.55 0.231 1.071|21.80 0.741 0.203|
> |OT-ODE|30.01 0.814 **0.077**|27.06 0.711 **0.126**|25.92 0.715 **0.109**|29.38 *0.839* 0.091|24.77 0.875 *0.085*|
> |PnP-Flow1|*31.17* *0.862* 0.136|*27.94* *0.759* 0.306|**26.96** **0.762** *0.170*|**33.01** **0.918** **0.037**|*26.46* *0.897* 0.102|
> |Flower1-OT(ours)|**31.66** **0.878** *0.104*|**28.63** **0.773** *0.255*|*26.24* *0.740* 0.273|*32.98* **0.918** *0.040*|**26.68** **0.915** **0.062**|
>
>
> > The overall improvements upon previous methods seem not very significant (in tab. 1 & 2).
>
> (b). While the numerical gains are not dramatic, they are consistent. Flower is the best or second-best method in most settings. Importantly, Flower achieves this while:
> - using nearly identical hyperparameters across all inverse problems,
> - providing a principled Bayesian derivation that connects PnP-style iterations with posterior sampling for flow-based priors.
>
> As the reviewer noted, this theoretical contribution is a key strength of the work.
>
> > In terms of the limitation of assumptions, i) the paper focuses only on linear problems, and ii) if I understand correctly the proximal step further assumes isotropic prior and measurement operation, which could be oversimplified.
>
> (c\). Beyond linearity, no additional assumption on the forward operator is imposed. To further validate this, we added experiments with Fourier sampling using different sampling masks (Appendix 8.5.7, Compressed Sensing Fourier Sampling).
>
> Regarding the noise model, our paper focused on isotropic Gaussian noise, which is a practical choice. Motivated by the reviewer’s remark, we have extended our theory to handle general non-isotropic Gaussian noise (Appendix 8.3) and included corresponding numerical examples (Appendix 8.5.7, Non-Isotropic Gaussian Noise).
>
> Finally, we added a discussion in Section 6 on potential extensions of Flower to nonlinear problems.
>
> > The most stand-out distinctions from previous flow-based inverse problem solvers are not very clear to me.
>
> (d). We revised Section 4 to clarify the distinctions. In summary:
>
> • OT-ODE estimates a new velocity field, whereas Flower adopts a Bayesian ancestral-sampling scheme that yields better measurement-consistent outputs.
>
> • D-Flow and Flow-Priors require costly backpropagation through ODE trajectories, while Flower avoids such computations and achieves better reconstruction quality.
>
> • PnP-Flow: Flower replaces the gradient-based update with a proximal refinement, which yields sharper reconstructions. Flower also provides a Bayesian interpretation absent in PnP-Flow.
>
> Overall, Flower combines competitive performance with a novel ancestral-sampling Bayesian theory, while remaining computationally efficient with minimal hyperparameter tuning.
>
> > Admittely, I am not very familiar with the current progress of this particular sub-domain. I would therefore also like to hear other colleague reviewers' opinions.
>
> (e). We are grateful the reviewer acknowledges this; please let us know if any additional clarification is needed.

---

> > ### Comment · Reviewer_kQup · 2025-11-26
> > **Thank you!**
> >
> > I appreciate the great efforts authors have made to address my concerns. I believe it further improves the quality of this manuscript and therefore would like to increase my original rating to 6.

---

### Author Response · Authors · 2025-11-20
**Summary of Updates to the Manuscript**

We want to thank all the reviewers once again for their valuable comments. Their feedback has been constructive, and we revised and uploaded the manuscript. In the updated version, we have:

- Provided results on larger-scale test sets for comparison of the methods in Appendix 8.5.6,
- Added further experiments on the application of Flower to Fourier sampling and non-isotropic Gaussian noise in Appendix 8.5.7,
- Included ablation studies on the effects of the number of evaluations for averaging in Appendix 8.5.4 and time discretization on the performance of Flower in Appendix 8.5.5,
- Added a discussion on the potential extension of Flower to nonlinear inverse problems in Section 6,
- Improved the overall clarity of the writing and corrected minor notational inconsistencies.

For convenience, we have uploaded a version of the paper with major changes highlighted in red as supplementary material. We remain available until the end of the discussion period for any further clarification.

---

### Author Response · Authors · 2025-12-04
**Summary for the Area Chair**

We thank the reviewers for their constructive comments. While we regret that the discussion could not be fully completed, we sincerely appreciate the organizers’ efforts in maintaining a fair rebuttal process. For documentation, here is a brief summary of the present state of affairs.

The reviewers highlighted Flower’s solid theoretical foundation, practical efficiency, and strong empirical performance for linear inverse problems, but also expressed concerns that were all answered in our responses. Specifically,
- ⁠For Reviewer Kgmh, we added the requested ablation studies, discussed potential nonlinear extensions of Flower, and addressed their questions regarding higher-order and adaptive solvers; the reviewer increased their confidence and maintained strong support for acceptance.
- ⁠For Reviewer rzV5, we adapted the writing of the paper to clarify the scope of the method, expanded the implementation details, and added larger-scale experiments; no further questions were raised.
- ⁠For Reviewer kQup, we added larger-scale evaluations along with non-isotropic noise and Fourier sampling experiments, and clarified distinctions from prior methods as well as the (minimal) assumptions required on the forward operator; the reviewer stated that these updates significantly improved the paper and increased their score.

We appreciate the AC’s effort in this cycle and look forward to the meta-review and final evaluation of our work.

---

### Meta-Review · Area_Chair_rFQQ · 2026-01-07

**Summary:**

This paper proposes a new flow-based approach for solving linear inverse problems. The method consists of three steps: (1) destination estimation, (2) projection, and (3) re-noising. The authors provide a solid theoretical analysis demonstrating the benefits of the approach, along with thorough experimental validation showing state-of-the-art results across various linear inverse problems. The ideas presented are interesting, and reviewers appreciated the technical soundness, theoretical contributions, and clear organization of the paper. The main concerns raised by reviewers were primarily related to experimental validation, all of which were sufficiently addressed in the rebuttal. Overall, the paper makes a valuable contribution to the field, and I recommend its acceptance.

**Reviewer Concerns:**

Reviewers expressed concerns about the empirical validation of the proposed approach, specifically regarding the marginal improvements offered by the algorithm, the small number of test images, and the limited ablation studies. All these concerns were addressed by the authors in their rebuttal by providing explanations and additional experiments.

**Reviewer Scores:**

Reviewer kQup initially gave a score of 4 and expressed an intention to increase it to 6 after the rebuttal. Reviewer rzV5 started with a score of 6, and given that all their comments were addressed, it is likely they would have raised their score. Finally, reviewer Kgmh gave a high score of 8 and indicated they would maintain it based on the discussion with the authors following the rebuttal.

---

### Decision · Program_Chairs · 2026-01-26

Accept (Poster)